# Time-Stepping FEM-Based Multi-Level Coupling of Magnetic Field–Electric Circuit and Mechanical Structural Deformation Models Dedicated to the Analysis of Electromagnetic Actuators

**Faiza Abba and M'hemed Rachek ***

Département d'Electrotechnique, Mouloud Mammeri University, BP 15000 Tizi-Ouzou, Algeria; aba-faiza@hotmail.com

\* Correspondence: rachek_mhemed@yahoo.fr; Tel.: +213-779-469-170

**Abstract:** The present paper introduced a framework for multi-level coupling transient electromagnetic fields (EMF) and mechanical structural dynamics based on the finite element method (FEM). This framework was dedicated to predicting, with better accuracy, the wave magnetic force density for obtaining the mechanical deformation occurring in electromagnetic actuators (EMAs). The first-level EMF transient model coupling is related to the magnetic field equations that are strongly coupled with the electric circuit input voltage equations. This is done by considering the magnetic saturation through the Newton–Raphson (N–R) method. The time-stepping solution of the EMF model resulted in the magnetic force densities being computed from the Lorentz force (LZ) expressions, based on the space–time variation of the induced eddy current. For the second coupling level, the EMF model was sequentially coupled with the mechanical structural deformation equations (MDef) through the local magnetic force density to achieve minimal material dynamic displacement and deformation. The developed multi-physics EMF–MDef time-stepping (FEM) model tools were implemented using the Matlab software.

**Keywords:** electromagnetic actuators; finite element method (FEM); transient electromagnetic model; multi-physics coupling; structural mechanical deformations

## 1. Introduction

Electromagnetic devices/actuators (EMDs/EMAs), such as electrical machines, sensors, actuators, magnetic/electrostatic micro- and nanoelectromechanical systems (MEMS/NEMS), etc., are currently used in a wide variety of applications ranging from industrial robotics/aerospace to automotive systems and biomedical devices that require high thrust, high accuracy, motion control, and different working ranges. The increased use of electromagnetic actuators raises the need for more improvements in the electromagnetic actuators, especially in terms of the unmatched combination of speed, precision, output force, and scalability. The operating principals of EMDs/EMAs are based on the interactions between the electromagnetic and mechanical structural dynamic phenomenon in weak couplings [1–5]. This phenomenon consists of electromagnetic induction excitations based on the magnetic force density and the structure mechanical stress response. Using the multi-physics approach to develop effective and cheaper innovative devices, it is recommended to carry out numerical investigations on the various physical parameters and effects (i.e., electromagnetic fields (EMF), circuits, vibration, noise, and structural deformation) on the device under consideration. The multi-physics numerical analysis of electromagnetic devices is based on the development of modern theoretical aspects and approaches for use in investigations in the industry [6,7].

Electromagnetic devices and actuators (EMDs/EMAs) are used in a wide variety of applications due to their electro–magneto–mechanical features. They are generally used to provide magnetic force and/or motion to position parts or close switches by transforming electrical signals to the linear motion of a moving armature. The prediction of the dynamic characteristics of electromagnetic actuators is a problem that involves the modeling of different mutually dependent domains as mechanical, structural, and electromagnetic, which are strongly influenced by their shape, material properties, and electric and mechanical elements. Therefore, in order to ensure a fast and efficient design, it is important to consider the finite element method (FEM) and simulations enabling the virtual prototyping of electromagnetic actuators over other available methods, such as the analytic or circuit equivalent methods [8–11]. A large magnetic force density is undesirable, since it generates vibration, acoustic noise, mechanical deformation (or displacement), disturbances in the electromagnetic devices, and material-based magnetostriction sheets. This can impede the system's performance [12,13]. The main part of the computational chain is the electromagnetic field simulation from which the surface force density waves are derived to obtain accurate mechanical deformation prediction. Accurate simulation plays an important role in the structure design, safety operation, and life time.

Evaluation of the magnetic force density effects in low magnetic fields is not overly complicated, and the existing techniques (i.e., Virtual Work, Maxwell Stress Tensor, or magnetic equivalent charges, etc.) implemented in available professional codes provide results with sufficient accuracy [13–17]. The problem of an electrically conducting object placed in a transient magnetic field is well known. Based on the Faraday law, when an electrically conducting object is placed across a variable magnetic field, an electromotive force (EMF) is induced inside the material. According to Ohm's law, it further produces eddy currents which follow circular paths in the planes normal to the field lines. However, in strong or pulsed magnetic fields, their determination is still a challenge. A Lorentz force eddy current (LZEC) characterizing linear motion and deformation of EMAs was built to exhibit a highly non-linear current for force relation, even when the conducting object was completely immersed in the highly non-linear transient magnetic field.

The aim of this research was to present the implementation of an electromagnetic–structural mechanic FEM-based equation model for the modeling of electromagnetic actuators in transient operating conditions, considering the non-linearity (NL) of the magnetic materials, as depicted by Figure 1. The generalized model focuses on the strong coupling between the partial differential equation of the magnetic field diffusion equation expressed in terms of the magnetic vector potential (MVP) and the electric circuit equations of the voltage-fed windings obtained from Kirchhoff laws. In addition, the EMF model integrates realistic geometries and the non-linear magnetic material properties through the magnetic flux density–magnetic permeability dependence handled by the iterative Newton–Raphson (N–R) method. The FEM formulation of the time-stepping non-linear coupled magnetic field electric circuit's two-dimensional model leads to a transient algebraic differential equations system. The solution process requires a major loop concerning the time-discretization using the effectiveness of the step-by-step numerical integration scheme, and then, for each time step, we have to ensure the minor loop convergence of the Newton–Raphson (N–R) algorithm for determining the appropriates magnetic permeability values.

Moreover, the structural mechanical deformation equation is sequentially coupled to the electromagnetic phenomenon through the magnetic force density to obtain the deformations. The electromagnetic model provides the normal (*y*-components) and tangential (*x*-components) components of the volume force density used as excitation for the structure–dynamic model, which allows the analysis of the mechanical deformation for small and large air gaps, electrical conductivity, and electrical voltage excitation for non-linear magnetic material properties.

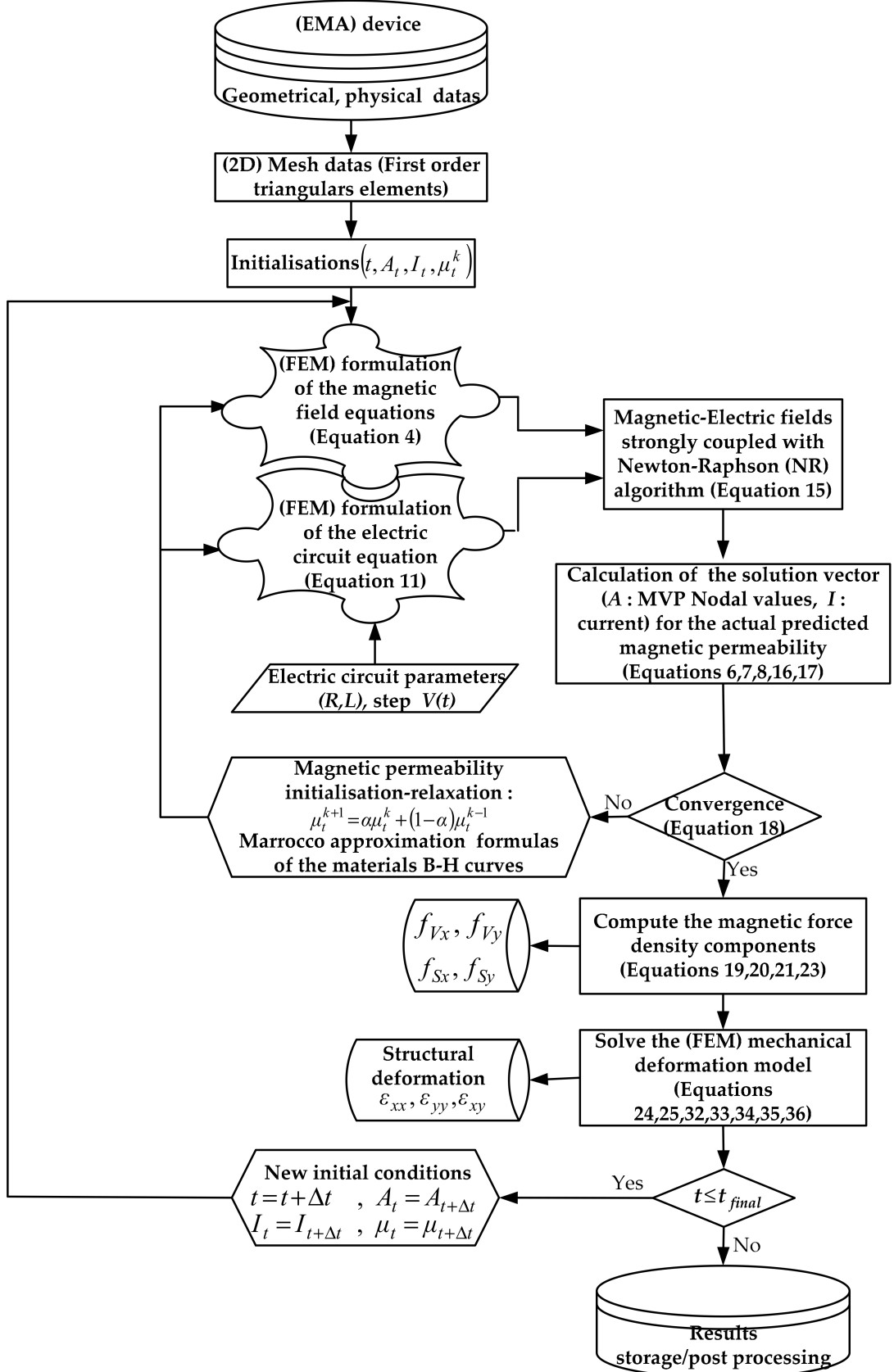

**Figure 1.** Flowchart of the magnetic field–electric circuit and structural mechanical multi-physical interactions.

The time variation and distribution of the magnetic force density was performed using the Lorentz eddy current (LZEC) formulas, based on the induced eddy currents and transient non-linear magnetic fields of the plate. This magnetic force density excitation was exported to the structural dynamic mesh of an EMA to compute the mechanical structural deformations.

## 2. Presentation of the Basic Electromagnetic Actuator

The schematic structure of the proposed electromagnetic actuator is illustrated in Figure 2. It consists of the voltage-fed winding electrical source $V_c(t)$. The winding $\Omega_{coil}$ regions with circulating current $I_c(t)$ domains are mounted on the ferromagnetic fixed core $\Omega_{core}$. The load region $\Omega_{load}$ is made of high-performance ferromagnetic material. The device components with length ($l_z$) are surrounded by an air-box region $\Omega_{air}$.

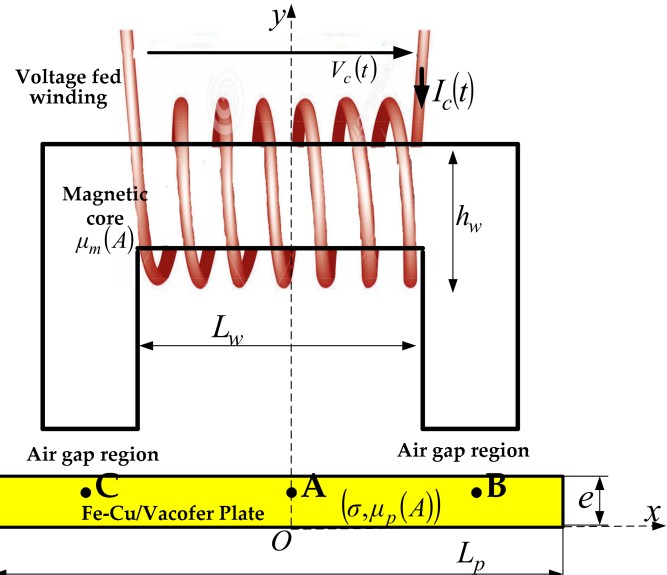

**Figure 2.** Typical components of the electromagnetic actuator.

The two-dimensional geometry is covered by a finite element mesh, consisting of first-order triangular elements generated by the MATLAB Partial Differential Equation (PDE) toolbox. This toolbox provides the ability to create a mesh using the Delaunay triangulation algorithm. The EM and structural mechanical models use the same mesh.

## 3. Strongly Coupled Magnetic Field–Circuit Formulation

### 3.1. Magnetic Field FEM Formulation

The electromagnetic field model in the magneto-dynamic problems is based on Maxwell's equations and on the concept of magnetic vector potential (MVP). The derived magnetic field equation for the different parts of the device is given as follows:

$$\vec{rot}\left(\frac{1}{\mu\left(\vec{A}\right)}\cdot\vec{rot}\left(\vec{A}\right)\right) = \begin{cases} 0 & \Omega_{air} \\ -\sigma\frac{\partial\vec{A}}{\partial t} & \Omega_{load} \\ 0 & \Omega_{core} \\ \pm\frac{N_c}{S_c}I_c & \Omega_{coil} \end{cases} , \tag{1}$$

where $\vec{A}$ is the magnetic vector potential, $I_c$ and $S_c$ are the winding current and the total cross-sectional area of the winding turns, respectively, and $N_c$ is the number of turns.

The physical properties of the materials are the electric conductivity and the non-linear magnetic material permeability $\mu(A)$ associated with the B–H magnetization curve. For the windings made up of stranded conductors, the current density is considered uniform over the cross-section of the conductors. Hence, the effect of eddy currents is negligible. For massive conducting materials, the eddy current is represented by the term $\sigma(\partial A_z / \partial t)$. The problem becomes two-dimensional (2D) in the $(x, y)$ plane; therefore, the magnetic vector potential only has the $z$-direction component $\overrightarrow{A}(0, 0, A_z)$. The above 2D magnetic field diffusion Equation (1) is described by the following system of equations:

$$
\frac{\partial}{\partial x}\left(\frac{1}{\mu(A_z)}\frac{\partial A_z(x,y,t)}{\partial x}\right) + \frac{\partial}{\partial y}\left(\frac{1}{\mu(A_z)}\frac{\partial A_z(x,y,t)}{\partial y}\right)
= \begin{cases}
0 & \Omega_{air} & (\mu_o) \\
-\sigma\frac{\partial A_z}{\partial t} & \Omega_{load} & (\mu_o\mu_r) \\
0 & \Omega_{core} & (\mu_o\mu_r(A_z)) \\
\pm\frac{Nc}{S_c}I_c & \Omega_{coil} & (\mu_o)
\end{cases}.
\tag{2}
$$

FEM is then used to discretize the solution region of the problem, where the unknowns are approximated as a linear combination of suitable functions within the appropriate boundary conditions. The magnetic vector potential $A_z(x, y, t)$ within an element $e$ is approximated by

$$
A_z(x, y, t) = \sum_{j=1}^{N_{nodes}} \alpha_j(x, y) A_{z_j}(x, y, t),
\tag{3}
$$

where $N_{nodes}$ is the total number of the nodes of the mesh, and $\alpha_j$ is Galerkin's shape function of node $j = 1, \ldots, N_{nodes}$ associated with the magnetic vector potential $A_{zj}$.

The FEM discretization is within the weighted residual method, which transforms the governing partial derivative equations into a variable form. Applying Galerkin's method and approximation function in Equation (3) to Equation (2) leads to the integral discrete equations of the magnetic field problems according to the air, load, core, and coil regions.

$$
\sum_{i=1}^{N_{elements}} \iint_{\Omega^e} \frac{1}{\mu^e(A_z)}[(\overrightarrow{\nabla}_{xy}\alpha_i)(\overrightarrow{\nabla}_{xy}\alpha_j)]\{A_{z_j}\}d\Omega^e - \sum_{i=1}^{N_{elements}} \oint_{\Gamma^e} \frac{1}{\mu^e}\left(\frac{\partial A_{z_j}}{\partial n}\right)d\Gamma^e
$$

$$
= \begin{cases}
0 & \Omega_{air} \\
-\sum_{i=1}^{N_{elements}} \iint_{\Omega^e_{load}} \sigma(\alpha_i\alpha_j)\frac{\partial\{A_z\}}{\partial t}d\Omega^e_{load} & \Omega_{load} \\
0 & \Omega_{core} \\
\sum_{i=1}^{N_{elements}} \iint_{\Omega^e_{coil}} \pm\frac{Nc}{S_c}(\alpha_i)\{I_c\}d\Omega^e_{coil} &
\end{cases}.
\tag{4}
$$

The matrix form of the discrete Equation (4) is expressed as follows:

$$
[S(\mu(A_z))]\{A_z\} = \begin{cases}
0 & \Omega_{air} \\
-[T]\frac{\partial\{A_z\}}{\partial t} & \Omega_{load} \\
0 & \Omega_{core} \\
[D^s]\{I_c\} & \Omega_{coil}
\end{cases}.
\tag{5}
$$

Entries in these matrices are given by

$$
S^e_{ij}(\mu(A_z)) = \iint \frac{1}{\mu^e(A_z)}\left(\frac{\partial\alpha_i}{\partial x}\frac{\partial\alpha_j}{\partial x} + \frac{\partial\alpha_i}{\partial y}\frac{\partial\alpha_j}{\partial y}\right)d\Omega^e,
\tag{6}
$$

$$
T^e_{ij} = \iint_{\Omega^e_{load}} \sigma(\alpha_i\alpha_j)d\Omega^e_{load},
\tag{7}
$$

$$D_{ij}^e = \iint\limits_{\Omega_{coil}^e} \pm \frac{N_c}{S_c}(\alpha_i)d\Omega_{coil}^e, \tag{8}$$

where $S(\mu)$ is the magnetic stiffness matrix, $D$ is the coupled matrix of the voltage source in the winding's parts, and T is the mass matrix associated with the magnetic induction phenomena.

### 3.2. Electric Circuit (FEM) Formulation

The winding with $N_c$ conductors is regarded as an equivalent lumped electric circuit per turn. The EMF parameters associated with the windings with a cross-section $S_c$ turn were proposed to be the longitudinal resistance $R_c$ and the self-inductance $L_{end}$ as depicted in Figure 3. The electric equation of the winding current $I_c(t)$ with the voltage $V_c(t)$ was obtained from the Kirchhoff law as follows:

$$V_c(t) = R_c I_c(t) + L_{end}\frac{dI_c(t)}{dt} + e_c(t). \tag{9}$$

The induced electromotive force in the winding is

$$e_c(t) = N_s \left[ \sum_{m=1}^{N_c} \iint\limits_{(S_c^m)^+} \frac{l}{(S_c^m)^+}\frac{dA_z}{dt}dS_c^+ - \sum_{m=1}^{N_c} \iint\limits_{(S_c^m)^-} \frac{l}{(S_c^m)^-}\frac{dA_z}{dt}dS_c^- \right]. \tag{10}$$

Substituting Equation (10) into Equation (9), the branch equation is

$$V_c(t) = R_c I_c(t) + L_{end}\frac{dI_c(t)}{dt} + N_s \left[ \sum_{m=1}^{N_c} \iint\limits_{(S_c^m)^+} \frac{l}{(S_c^m)^+}\frac{dA_z}{dt}dS_c^+ - \sum_{m=1}^{N_c} \iint\limits_{(S_c^m)^-} \frac{l}{(S_c^m)^-}\frac{dA_z}{dt}dS_c^- \right]. \tag{11}$$

The sections $(S_c^+)$ and $(S_c^-)$ denote the positively and negatively oriented cross-sections of the coil, and $N_s$ is the number of symmetry sectors. The summation over the coil number concerns the positively and negatively oriented cross-sections of the coils located in the solution region $\Omega_{coil}$.

After the use of FEM discretization and the MVP approximation function in Equation (3), the FE formulation of the electric circuit Equation (12) was obtained.

$$V_c(t) = R_c I_c(t) + L_{end}\frac{dI_c(t)}{dt} + lN_s[D^s]^{tr}\frac{d\{A_z\}}{dt}. \tag{12}$$

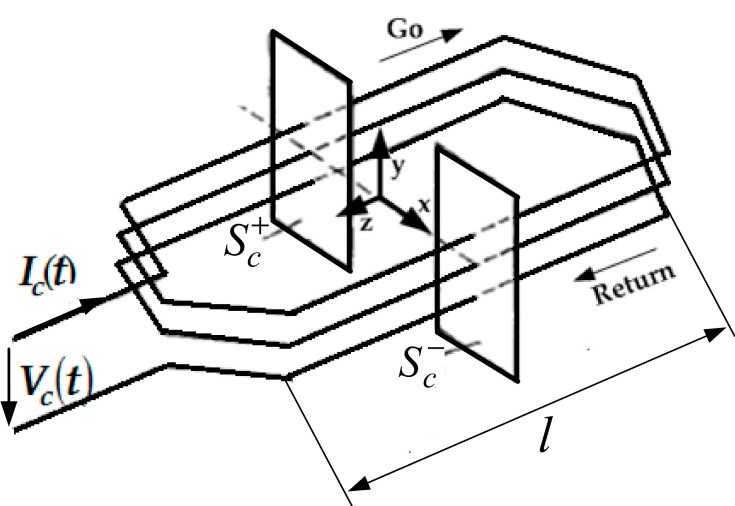

**Figure 3.** Winding electric-fed voltage circuit.

### 3.3. Non-Linear Time-Stepping Magnetic Field–Circuit Coupled Model

The electric circuit Equation (12) for the windings is directly coupled with the MVP-based magnetic fields Equation (5) in the cross-section of the actuator. The set of the strongly coupled equations leads to the differential first-order algebraic system of equations written as

$$
\begin{bmatrix} S(v) & -D^s \\ 0 & R_c \end{bmatrix} \begin{Bmatrix} A_z \\ I_c \end{Bmatrix} + \begin{bmatrix} [T] & 0 \\ lN_s[D^s]^{tr} & L_{end} \end{bmatrix} \begin{Bmatrix} \frac{dA_z}{dt} \\ \frac{dI_c}{dt} \end{Bmatrix} = \begin{Bmatrix} [A_z]_{CL}^{\Omega} \\ V_c(t) \end{Bmatrix},
\tag{13}
$$

where the unknowns $[A_z]$ and $[I_c]$ are the magnetic vector potentials and the winding current, respectively, that are required to be evaluated.

In Equation (13), the time derivatives of the vector potential and the winding currents are approximated by first-order difference ratios. However, there are basic methods of time discretization: the forward difference method ($\beta = 0$), backward difference method ($\beta = 1$), Crank–Nicholson method ($\beta = 1/2$), and the Galerkin method ($\beta = 3/2$) [18–20].

$$
\beta \frac{d}{dt} \begin{Bmatrix} A_z \\ I_c \end{Bmatrix}_{t+\Delta t} + (1-\beta)\frac{d}{dt} \begin{Bmatrix} A_z \\ I_c \end{Bmatrix}_t = \frac{\left( \begin{Bmatrix} A_z \\ I_c \end{Bmatrix}_{t+\Delta t} - \begin{Bmatrix} A_z \\ I_c \end{Bmatrix}_t \right)}{\Delta t}.
\tag{14}
$$

After having expressed Equation (13) of the unknown MVP $A_z$ and current $I_c$ for the $N$ nodes and $Nc$ conductors at time $t$ and $t + \Delta t$ using Equation (14), the algebraic equation system to be solved is given by the algebraic equation system in Equation (15) as follows:

$$
\left( \beta \begin{bmatrix} S(\mu(A_z)) & -D^s \\ 0 & R_c \end{bmatrix} + \frac{1}{\Delta t} \begin{bmatrix} [T] & 0 \\ lN_s[D^s]^{tr} & L_{end} \end{bmatrix} \right) \begin{Bmatrix} A_z \\ I_c \end{Bmatrix}_{t+\Delta t} = -\left( (1-\beta) \begin{bmatrix} S(v) & -D^s \\ 0 & R_c \end{bmatrix} - \frac{1}{\Delta t} \begin{bmatrix} [T] & 0 \\ lN_s[D^s]^{tr} & L_{end} \end{bmatrix} \right) \begin{Bmatrix} A_z \\ I_c \end{Bmatrix}_t
$$
$$
+ \begin{Bmatrix} [A_z]_{CL}^{\Omega} \\ \beta V_c(t+\Delta t) + (1-\beta) V_c(t) \end{Bmatrix}.
\tag{15}
$$

The solution of the algebraic equations system in Equation (15) needs to be carried out iteratively as a series of sequential linear tasks. The problem is non-linear due to the presence of ferromagnetic materials that have properties governed by the magnetization B–H curve. Here, the linearization is done based on the Newton–Raphson (N–R) method [21,22]. Extraction of the data along the curve herein uses the Morocco approximation formula [23,24]. For all points beyond the range of available data, the curve can be linearly extrapolated. At every iteration $(n + 1)$, a new estimate nodal value of the magnetic vector potential is obtained after correcting the inaccurate result of the previous iteration $(n)$.

The stiffness matrix $[P]$ depends on the nodal values of the magnetic vector potential. After applying the N–R iteration method, a final algebraic system of equations for the non-linear time-stepping simulation of the actuators is obtained as

$$
\left( \beta \begin{bmatrix} P\left(\mu\left(A_z^{k+1}\right)\right) + \frac{[T]}{\Delta t} & -D^s \\ \frac{lN_s[D^s]^{tr}}{\Delta t} & R_c + \frac{L_{end}}{\Delta t} \end{bmatrix} \right) \begin{Bmatrix} \Delta A_z \\ \Delta I_c \end{Bmatrix}_{t+\Delta t}^{k+1} = -\left( (\beta) \begin{bmatrix} S\left(\mu\left(A_z^k\right)\right) + \frac{[T]}{\Delta t} & -D^s \\ \frac{lN_s[D^s]^{tr}}{\Delta t} & R_c + \frac{L_{end}}{\Delta t} \end{bmatrix} \right) \begin{Bmatrix} A_z \\ I_c \end{Bmatrix}_{t+\Delta t}^{k}
$$
$$
+ \left( (1-\beta) \begin{bmatrix} S(\mu(A_z)) - \frac{[T]}{\Delta t} & -D^s \\ -\frac{lN_s[D^s]^{tr}}{\Delta t} & R_c - \frac{L_{end}}{\Delta t} \end{bmatrix} \right) \begin{Bmatrix} A_z \\ I_c \end{Bmatrix}_t + \begin{Bmatrix} [A_z]_{CL}^{\Omega} \\ \beta V_c(t+\Delta t) + (1-\beta) V_c(t) \end{Bmatrix},
\tag{16}
$$

where $[P]$ is the Jacobian matrix system, expressed through the following matrix elements [23]:

$$
P_{ij}^e = \iint_{\Omega^e} \left[ \frac{\partial}{\partial A_{zj}} \left( \frac{1}{\mu(A_{zj})} \right) \right] \left( \vec{\nabla} \alpha_i \bullet \vec{\nabla} \alpha_j \right) d\Omega^e.
\tag{17}
$$

To update the iterative values of the nodal magnetic vector potential, the relaxation $0.5 \leq \theta \leq 1$ factor may be used according to

$$\theta \begin{bmatrix} \{\Delta A_z\} \\ \{\Delta I_c\} \end{bmatrix}_{t+\Delta t}^{k+1} = \begin{bmatrix} \{A_z\} \\ \{I_c\} \end{bmatrix}_{t+\Delta t}^{k+1} - \begin{bmatrix} \{A_z\} \\ \{I_c\} \end{bmatrix}_{t+\Delta t}^{k}. \tag{18}$$

## 4. Magnetic Eddy Current Force Calculation

To evaluate the mechanical deformation, we need to know the distribution of the magnetic force density. There are several methods of formulating the volume and surface electromagnetic forces densities exerted on a ferromagnetic medium. These methods are based on different physical or mathematical representations of the ferromagnetic media. They share the same global force exerted throughout the ferromagnetic media, but postulate different volume $f_V$ and surface $f_S$ magnetic force densities. The Ampererian Force representation, Maxwell Stress Method, and Virtual Work method are considered in References [13,16,17,25]. The volume magnetic force density is given by the Maxwell Stress Tensor method based on the Lorentz Force formula expressed from the induced eddy currents and magnetic flux densities according to the non-linear magnetic properties of the interested plate. However, there is another kind of force called "magnetization force" which is caused by the changes in permeability. According to the Korteweg–Helmholtz force density law, a revised form of Lorentz force formula calculates the force density in a rigid body as

$$f^{em} = f_V^{em} + f_S^{em} = \left( \vec{J}_{eddy} \times \vec{B} \right) - \frac{1}{2} \left( H_t^2 + H_n^2 \right) \nabla \mu, \tag{19}$$

where $H_t$ and $H_n$ are the tangential and normal magnetic field located on the plate surrounding the surface. For 2D $(x, y)$ coordinates, $H_t = H_x$ and $H_n = H_y$.

The components of the volume magnetic force density components $(f_{Vx}, f_{Vy})$ expressed from the Lorentz eddy current force (LZEC) formulas in non-linear magnetic material (NL) can be written as follows for the 2D $(x, y)$ plane:

$$[f_V^{em}] = \mu(A_z) \left( \vec{J}_{eddy} \times \vec{H} \right) = \begin{bmatrix} f_{Vx} \\ f_{Vy} \end{bmatrix} = J_{eddy}^{t+\Delta t} \begin{bmatrix} \left( \frac{\partial A_z^e}{\partial x} \right) \vec{i} \\ \left( \frac{\partial A_z^e}{\partial y} \right) \vec{j} \end{bmatrix}^{t+\Delta t}, \tag{20}$$

where $J_{eddy}^{t+\Delta t}$ is the induced eddy current density, $\vec{H}$ is the magnetic field, and $A_z^e$ is the magnetic vector potential values of the finite element barycenter obtained from the nodal values.

According to the magnetic reluctivity at each time step, using the finite element method, the magnetic flux density is the first post-processed value computed from the barycenter values of the magnetic vector potential on each triangular element. The second post-processed value at each time step is the induced eddy current evaluated from the magnetic vector potential based on the time-discretization formula (21).

$$J_{eddy}^{t+\Delta t} = \frac{\beta - 1}{\beta} J_{eddy}^t - \sigma \left( \frac{(A_z^e)_{t+\Delta t} - (A_z^e)_t}{\beta \Delta t} \right). \tag{21}$$

Also, considering the application of continuity theorem in finite element calculation, final expression of the stress on a linear, homogeneous, isotropic, and non-compressible ferromagnetic material surface is given by

$$f_S^{em} = -\frac{1}{2} \left[ H_t^2 \left( \mu_o - \mu_p(A_z) \right) + B_n^2 \left( \frac{1}{\mu_p(A_z)} - \frac{1}{\mu_o} \right) \right]. \tag{22}$$

For high magnetic permeability as in a ferromagnetic body, the tangential component of the magnetic field outside the surface is near zero. Thus, the force is approximately normal to the surface and is found from the integral of the magnetic tension over the surface.

$$f_S^{em} = \begin{bmatrix} f_{Sx} \\ f_{Sy} \end{bmatrix} = \begin{bmatrix} 0 \cdot \vec{i} \\ -\frac{1}{2}\left(\frac{1}{\mu_p(A_z)} - \frac{1}{\mu_o}\right)B_y^2 \cdot \vec{j} \end{bmatrix} = \begin{bmatrix} 0 \cdot \vec{i} \\ \frac{1}{2}\left(\frac{1}{\mu_o} - \frac{1}{\mu_p(A_z)}\right)\left(\frac{\partial A_z}{\partial x}\right)^2 \cdot \vec{j} \end{bmatrix}. \tag{23}$$

From Equation (23), the amplitude of $f_S^{em}$ only depends on the strength of the normal magnetic field and the shape of the surface surrounding the plate.

## 5. Mechanical Deformation FEM Formulation Model

In electromagnetic–mechanical devices, the magnetic materials are subject to displacements and deformations under the action of the distributed magnetic force density [25–27]. To study the problem, the weakly coupled model between the magnetic–electric circuit and the structural mechanical fields is required. After the electromagnetic simulation, a structural dynamic simulation is performed to determine the deformations components. The magnetic force density obtained from the electromagnetic simulations in the case of non-linear magnetic properties is used as the excitation of the FEM formulation structural dynamic simulation model. The mechanical analysis predicts the mechanical stress and deformation of the actuators. This is governed by the compatibility equation, the constitution equation, and the equilibrium equation.

### 5.1. Equilibrium Equations

Interactions of electromagnetic fields in ferromagnetic media induce volume $f_V$ and surface $f_S$ electromagnetic force density sources occurring in the mechanical equilibrium equations. The static mechanical equilibrium equations when assuming small deformations and non-dynamic behavior due to the inertia can be written as

$$\nabla \bullet \sigma(U) + f_V = 0 \ \Omega_{load}, \tag{24}$$

$$\sigma(U) \bullet n = f_S \ \Gamma_{load}, \tag{25}$$

where $\sigma(U)$ is the mechanical stress tensor, $U$ is the displacement vector, and fem/tem are the volume and surface external electromagnetic force density field, respectively. Since the magnetostriction phenomena resulting from the strong magneto-mechanical elastic coupling is not taken into account, the only applied forces are the magnetic forces densities computed from Equations (22) and (23). In addition, there are no moments proportional to a volume, which is the case for most solid materials, where the stress tensor is symmetric.

According to (2D) $(x, y)$ Cartesian coordinate system, the structural mechanical equilibrium differential equation expresses the relationship between the mechanical stress components and the magnetic force density [28,29].

$$\begin{cases} \frac{\partial \sigma_{xx}}{\partial x} + \frac{\partial \sigma_{xy}}{\partial y} + f_{Vx} = 0 \\ \frac{\partial \sigma_{xy}}{\partial x} + \frac{\partial \sigma_{yy}}{\partial y} + f_{Vy} = 0 \end{cases}, \tag{26}$$

where $\sigma_{xx}, \sigma_{yy}, \sigma_{xy}$ denote the stresses along the $x$- and $y$-directions, respectively, and $f_{Vx}, f_{Vy}$ are the body magnetic force densities acting along the $x$- and $y$-directions, respectively.

To compute the displacement caused by an applied load, a relationship between stress $\sigma(U)$ and displacement $U$ is required. This is done in two steps. Firstly, the strain tensor $\varepsilon$ is determined as a function of $U$, and then the stress is determined as a function of the strain. These relationships are, in principle, non-linear; however, once it is assumed that the stresses and strains remain relatively small, the relationships can be assumed to be linear, which yields the linear relationships detailed in the next section.

### 5.2. Constitution Equation (Stress–Strain)

From the concepts of stress and strain, the generalized Hook's law states that the components of stress are linearly related to the components of strain [30]. The generalized stress $\{\sigma\}$–strain $\{\varepsilon\}$

relationship given by Hook's law in the case of a linear elastic isotopic two-dimensional solid is written as

$$\{\sigma\}^T = \left\{ \begin{array}{c} \sigma_{xx} \\ \sigma_{yy} \\ \sigma_{xy} \end{array} \right\} = \left[ \begin{array}{ccc} G_{11} & G_{12} & 0 \\ G_{21} & G_{22} & 0 \\ 0 & 0 & G_{33} \end{array} \right] \left\{ \begin{array}{c} \varepsilon_x \\ \varepsilon_y \\ \varepsilon_{xy} \end{array} \right\} = [G]\{\varepsilon\}, \tag{27}$$

where $G_{ij}$ is the reduced stiffness coefficient, given by

$$G_{11} = G_{22} = \frac{E}{1-v^2}, G_{12} = G_{21} = \frac{Ev}{1-v^2}, G_{33} = \frac{E(1-v)}{2(1-v^2)}. \tag{28}$$

The material parameters E and $v$ are the Young's modulus and Poisson's ratio, respectively.

### 5.3. Compatibility Equation (Strain–Displacement)

The deformed shape of an elastic body under any given two-dimensional device can be completely described by the two components of the independent displacements $u$ and $v$, which are parallel to the $x$- and $y$-directions, respectively. In general, each of these components $u$ and $v$ is a function of the Cartesian coordinates $x$ and $y$. According to the consideration of small deformations, the linear strain deformation–displacement relationship is expressed in general matrix form as follows:

$$\{\varepsilon\}^T = \left\{ \begin{array}{c} \varepsilon_{xx} \\ \varepsilon_{yy} \\ \varepsilon_{xy} \end{array} \right\} = \left\{ \begin{array}{c} \frac{\partial u(x,y)}{\partial x} \\ \frac{\partial v(x,y)}{\partial y} \\ \frac{\partial u(x,y)}{\partial y} + \frac{\partial v(x,y)}{\partial x} \end{array} \right\}. \tag{29}$$

The combining of Equations (26), (27), and (29) has eight unknowns (i.e., three stresses, three strains, and two displacements) for eight equations (two equilibrium, three constitutive, and three kinematic equations).

### 5.4. Finite Element Formulations

To develop the finite element formulation for the mechanical deformation problem, we can apply Galerkin's method. Applying the weighted residual method to Equation (26) and applying the substitution of the constitutive equation (stress–strain) gives us

$$\int_{\Omega_{load}} \left\{ \begin{array}{c} \psi_1 \left( \frac{\partial \sigma_{xx}}{\partial x} + \frac{\partial \sigma_{xy}}{\partial y} \right) \\ \psi_2 \left( \frac{\partial \sigma_{yx}}{\partial x} + \frac{\partial \sigma_{yy}}{\partial y} \right) \end{array} \right\} d\Omega_{load} - \int_{\Omega} \left\{ \begin{array}{c} \psi_1 f_{Vx} \\ \psi_2 f_{Vy} \end{array} \right\} d\Omega_{load} - \int_{\Gamma_{load}} \left\{ \begin{array}{c} \psi_1 f_{Sx} \\ \psi_2 f_{Sy} \end{array} \right\} d\Gamma_{load} = 0, \tag{30}$$

where $\Omega_{load}$ is the mechanical domain of the plate, $\Gamma_{load}$ is the boundary of $\Omega_{load}$, $\psi_1$ and $\psi_2$ are the weighting functions, and $f_{Sx}, f_{Sy}$ are the surface magnetic force densities acting along the boundary.

Applying integration by parts to the terms of the first integral in Equation (30) yields:

$$\int_{\Omega_{load}} \left[ \begin{array}{ccc} \frac{\partial \psi_1}{\partial x} & 0 & \frac{\partial \psi_1}{\partial y} \\ 0 & \frac{\partial \psi_2}{\partial y} & \frac{\partial \psi_2}{\partial x} \end{array} \right] \left\{ \begin{array}{c} \sigma_{xx} \\ \sigma_{yy} \\ \sigma_{xy} \end{array} \right\} d\Omega_{load} = \int_{\Omega_{load}} \left\{ \begin{array}{c} \psi_1 f_{Vx} \\ \psi_2 f_{Vy} \end{array} \right\} d\Omega_{load} + \int_{\Gamma_{load}} \left\{ \begin{array}{c} \psi_1 f_{Sx} \\ \psi_2 f_{Sy} \end{array} \right\} d\Gamma_{load}. \tag{31}$$

After substitution of the stress–tensor Equation (27) and the strain–displacement Equation (29) into Equation (31), this gives

$$\int_{\Omega_{load}} \left[ \begin{array}{ccc} \frac{\partial \psi_1}{\partial x} & 0 & \frac{\partial \psi_1}{\partial y} \\ 0 & \frac{\partial \psi_2}{\partial y} & \frac{\partial \psi_2}{\partial x} \end{array} \right] [G] \left\{ \begin{array}{c} \frac{\partial u}{\partial x} \\ \frac{\partial v}{\partial y} \\ \frac{\partial u}{\partial y} + \frac{\partial v}{\partial x} \end{array} \right\} d\Omega_{load} = \int_{\Omega_{load}} \left\{ \begin{array}{c} \psi_1 f_{Vx} \\ \psi_2 f_{Vy} \end{array} \right\} d\Omega_{load} + \int_{\Gamma_{load}} \left\{ \begin{array}{c} \psi_1 f_{Sx} \\ \psi_2 f_{Sy} \end{array} \right\} d\Gamma_{load}. \tag{32}$$

Let us discretize the domain using linear triangular elements. Then, both displacements $u$ and $v$ are interpolated using the same shape functions as

$$
\left\{ \begin{array}{c} u(x,y) \\ v(x,y) \end{array} \right\} = \left\{ \begin{array}{c} \sum\limits_{j=1}^{3} \alpha_j(x,y) u_j \\ \sum\limits_{j=1}^{3} \alpha_j(x,y) v_j \end{array} \right\} = \left[ \begin{array}{cccccc} \alpha_1(x,y) & 0 & \alpha_2(x,y) & 0 & \alpha_3(x,y) & 0 \\ 0 & \alpha_1(x,y) & 0 & \alpha_2(x,y) & 0 & \alpha_3(x,y) \end{array} \right] \left\{ \begin{array}{c} u_1 \\ v_1 \\ u_2 \\ v_2 \\ u_3 \\ v_3 \end{array} \right\} = [\,]\{U\}, \quad (33)
$$

where $\alpha_j(x,y)$ is the shape function associated with the displacements of the nodes $j = 1,2,3$ of each triangular element $j$. The use of Equation (33) for the strain Equation (29) yields

$$
\left\{ \begin{array}{c} \frac{\partial u}{\partial x} \\ \frac{\partial v}{\partial y} \\ \frac{\partial u}{\partial y} + \frac{\partial v}{\partial x} \end{array} \right\} = \left[ \begin{array}{cccccc} \frac{\partial \alpha_1}{\partial x} & 0 & \frac{\partial \alpha_2}{\partial x} & 0 & \frac{\partial \alpha_3}{\partial x} & 0 \\ 0 & \frac{\partial \alpha_1}{\partial y} & 0 & \frac{\partial \alpha_2}{\partial y} & 0 & \frac{\partial \alpha_3}{\partial y} \\ \frac{\partial \alpha_1}{\partial y} & \frac{\partial \alpha_1}{\partial x} & \frac{\partial \alpha_2}{\partial y} & \frac{\partial \alpha_2}{\partial x} & \frac{\partial \alpha_3}{\partial y} & \frac{\partial \alpha_3}{\partial x} \end{array} \right] \left\{ \begin{array}{c} u_1 \\ v_1 \\ u_2 \\ v_2 \\ u_3 \\ v_3 \end{array} \right\} = [B]\{U\}. \quad (34)
$$

The final discrete integral form of the mechanical deformation Equation (32) after substituting Equation (29) is obtained from the FEM formulation of each triangular element as follows:

$$
\int\limits_{\Omega_{load}} \left[ \begin{array}{ccc} \frac{\partial \psi_1}{\partial x} & 0 & \frac{\partial \psi_1}{\partial y} \\ 0 & \frac{\partial \psi_2}{\partial y} & \frac{\partial \psi_2}{\partial x} \end{array} \right] [G] \left[ \begin{array}{cccccc} \frac{\partial \alpha_1}{\partial x} & 0 & \frac{\partial \alpha_2}{\partial x} & 0 & \frac{\partial \alpha_3}{\partial x} & 0 \\ 0 & \frac{\partial \alpha_1}{\partial y} & 0 & \frac{\partial \alpha_2}{\partial y} & 0 & \frac{\partial \alpha_3}{\partial y} \\ \frac{\partial \alpha_1}{\partial y} & \frac{\partial \alpha_1}{\partial x} & \frac{\partial \alpha_2}{\partial y} & \frac{\partial \alpha_2}{\partial x} & \frac{\partial \alpha_3}{\partial y} & \frac{\partial \alpha_3}{\partial x} \end{array} \right] \left\{ \begin{array}{c} u_1 \\ v_1 \\ u_2 \\ v_2 \\ u_3 \\ v_3 \end{array} \right\} d\Omega_{load}^e
$$

$$
= \int\limits_{\Omega_{load}^e} \left\{ \begin{array}{c} \psi_1 f_{Vx} \\ \psi_2 f_{Vy} \end{array} \right\} d\Omega_{load}^e + \int\limits_{\Gamma_{load}^e} \left\{ \begin{array}{c} \psi_1 f_{Sx} \\ \psi_2 f_{Sy} \end{array} \right\} d\Gamma_{load}^e. \quad (35)
$$

Upon applying the well-known Galerkin's method, which states the weighted functions as following $\Psi_1 = \alpha_j (j = 1,2,3)$ and $\Psi_2 = \alpha_j (j = 1,2,3)$ for each mesh element of the discretized domain, Equation (35) becomes

$$
\sum\limits_{i=1}^{N_{element}^{load}} \left[ \begin{array}{cc} \int\limits_{\Omega_{load}^e} \left( \frac{\partial \psi_i}{\partial x} G_{11} \frac{\partial \alpha_j}{\partial x} + \frac{\partial \psi_i}{\partial y} G_{33} \frac{\partial \alpha_j}{\partial y} \right) & \int\limits_{\Omega_{load}^e} \left( \frac{\partial \psi_i}{\partial x} G_{12} \frac{\partial \alpha_j}{\partial y} + \frac{\partial \psi_i}{\partial y} G_{33} \frac{\partial \alpha_j}{\partial x} \right) \\ \int\limits_{\Omega_{load}^e} \left( \frac{\partial \psi_i}{\partial x} G_{21} \frac{\partial \alpha_j}{\partial y} + \frac{\partial \psi_i}{\partial y} G_{33} \frac{\partial \alpha_j}{\partial x} \right) & \int\limits_{\Omega_{load}^e} \left( \frac{\partial \psi_i}{\partial y} G_{22} \frac{\partial \alpha_j}{\partial y} + \frac{\partial \psi_i}{\partial x} G_{33} \frac{\partial \alpha_j}{\partial x} \right) \end{array} \right] \left\{ \begin{array}{c} u_j \\ v_j \end{array} \right\} d\Omega_{load}^e
$$

$$
= \sum\limits_{i=1}^{N_{element}^{load}} \left( \int\limits_{\Omega_{load}^e} \left[ \begin{array}{cc} \psi_i & 0 \\ 0 & \psi_i \end{array} \right] \left\{ \begin{array}{c} f_{Vxj} \\ f_{Vyj} \end{array} \right\} d\Omega_{load}^e + \int\limits_{\Gamma_{load}^e} \left[ \begin{array}{cc} \psi_i & 0 \\ 0 & \psi_i \end{array} \right] \left\{ \begin{array}{c} f_{Sxj} \\ f_{Syj} \end{array} \right\} d\Gamma_{load}^e \right), \quad (36)
$$

where $N_{element}^{load}$ and $\Omega_{load}^e$ denote the number of the triangular element of the ferromagnetic plate and the elementary finite element domain. As a result, the stiffness matrix [K] and magnetic force density vector [F] build the global algebraic equations system written as

$$
[K]\{U\} = [F_V] + [F_S] = [F]. \quad (37)
$$

According to Equation (36), the elementary stiffness matrix and source vector components of the algebraic system in Equation (37) are expressed as follows:

$$[K_{ij}] = \int\limits_{\Omega^e_{load}} [B]^T [G][B] d\Omega^e_{load}. \tag{38}$$

## 6. Application, Results, and Discussion

In this section, we present the results of the simulations obtained from the computation code analysis package developed and implemented under the Matlab environment. This code was based on the finite element method (FEM), adopted as a method of partial-derivative discretization describing the multi-level coupling between the electric–magnetic fields and mechanical deformation phenomena, as depicted by Figure 1. The first level was a strong coupling of the transient magnetic equations and the electrical circuit, while the magnetic non-linear material was the use of the Newton–Raphson algorithm (N–R). The second-level coupling was ensured by the magnetic force density computed from the Lorentz eddy current formula. This is the weak coupling between the electromagnetic problem and the structural–mechanical equations of the deformations. To perform the deformations as accurately as possible, it was essential to accurately compute the transient and highly non-linear distribution of the magnetic forces and the induced eddy currents. Magnetic non-linearity (NL) is a fundamental phenomenon because this effect modulates the magnitude and distribution of the magnetic flux density.

At each time step, an electromagnetic field solution was calculated, followed by a structural displacement field solution. The simulated actuator had parts of saturable magnetic material, which implied a non-linear permeability. Magnetic boundary conditions were applied to the outer edge of the model. They forced the magnetic flux to be tangential to the model boundary, confining the flux within the model. Structural boundary conditions were specified on every element.

This numerical model was applied to the electromagnetic actuator given in Figure 2. The relevant geometrical, electrical, and mechanical parameters of the electromagnetic actuators are listed in Table 1; Table 2.

**Table 1.** Geometrical parameters of the actuators.

| Parameters | Plate Length (Lp) | Winding Width (Hw) | Winding Length (Lw) | Plate Thickness (e) | Air-Gap Thickness |
|---|---|---|---|---|---|
| Value (mm) | 90 | 5 | 15 | 7 | 1–5 |

**Table 2.** Mechanical and electrical parameters of the actuators.

| Parameters | Young's Modulus (E) | Poisson Ratio (υ) | Winding Resistance (Rc) | Winding Inductance (L_end) | Plate Electrical Conductivity (unit MS/m) |
|---|---|---|---|---|---|
| Value | 200 kN/mm$^2$ | 0.24 (Fe–Cu alloy) 0.33 (Vacofer S1) | 1 Ω | 5 mH | 9.1 (Fe–Cu alloy) 10.21 (Vacofer S1) |

Both homogeneous Dirichlet and Neumann boundary conditions are applied according to electromagnetic and mechanical deformation models as depicted in Figure 4.

The mechanical deformation constraints are detailed in Figure 4. Homogeneous Dirichlet boundary conditions are imposed for the displacement components (*u* and *v*) according to the clamped-clamped left and right sides of the plate. On the other hand, homogeneous Neumann boundary conditions with free displacement components are imposed on both horizontal boundaries. In addition, the constraints associated with the surface magnetic force density are also established through Equation (36), particularly on the plate boundary facing the ferromagnetic core and coil.

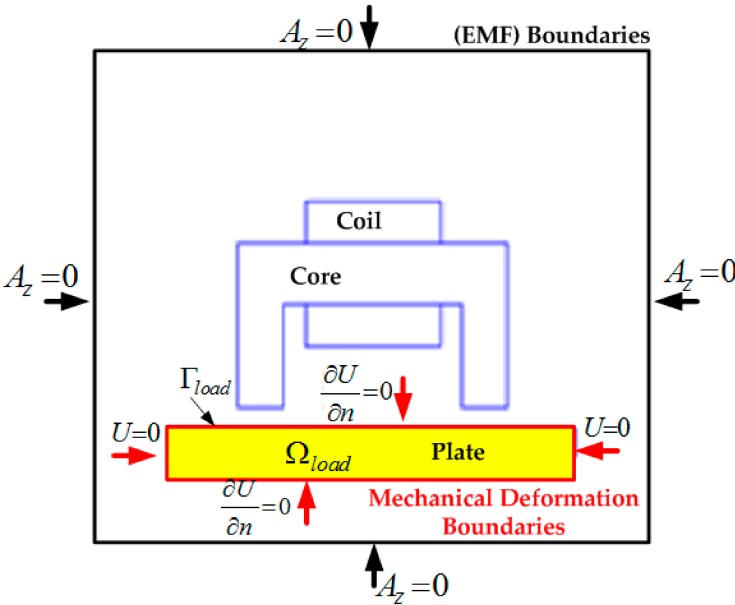

**Figure 4.** Electromagnetic and mechanical deformation boundary conditions.

### 6.1. The Results of the Electromagnetic Simulations (FEM)

The results of this section concern the 2D FEM transient electromagnetic simulation. The finite element mesh obtained using the automatic mesh generator from the Matlab PDE tool package contained 8476 first-order triangular elements and 4259 nodes. The winding was fed by a step voltage. The simulation duration was 50 ms, with a time step of Δt = 1 ms. The conducting plate domain associated with the mechanical deformation problem contained 3584 triangular elements. At each step time, the algebraic system in Equation (16) corresponding to the Newton–Raphson algorithm was iteratively solved to obtain the permeability value. The latest value was then used to establish the algebraic system in Equation (15), where the solution led to the node values of the magnetic vector potential and the winding current at each time step.

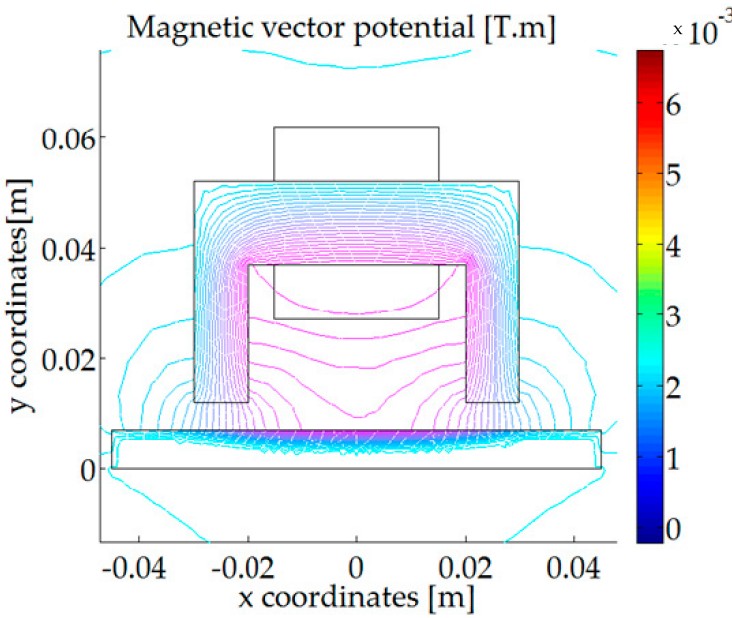

**Figure 5.** Steady-state distribution of the magnetic vector potential.

Figure 5 shows the equipotential lines of the steady-state magnetic vector potential nodal values, which are particularly concentrated on the cross-section of the plate facing the magnetic core and excited winding.

The field-line distribution and the magnetic flux density map vectors are plotted in Figure 6. We note a significant magnetic flux passing through the cross-section of the plate facing the magnetic core. This would imply a high amount of forces being generated throughout the plate due to eddy currents. The maximum value of the magnetic flux density in the plate domain was about 1.75 T, which corresponded to the non-linear region of the B–H curve associated with the Vacofer S1 material.

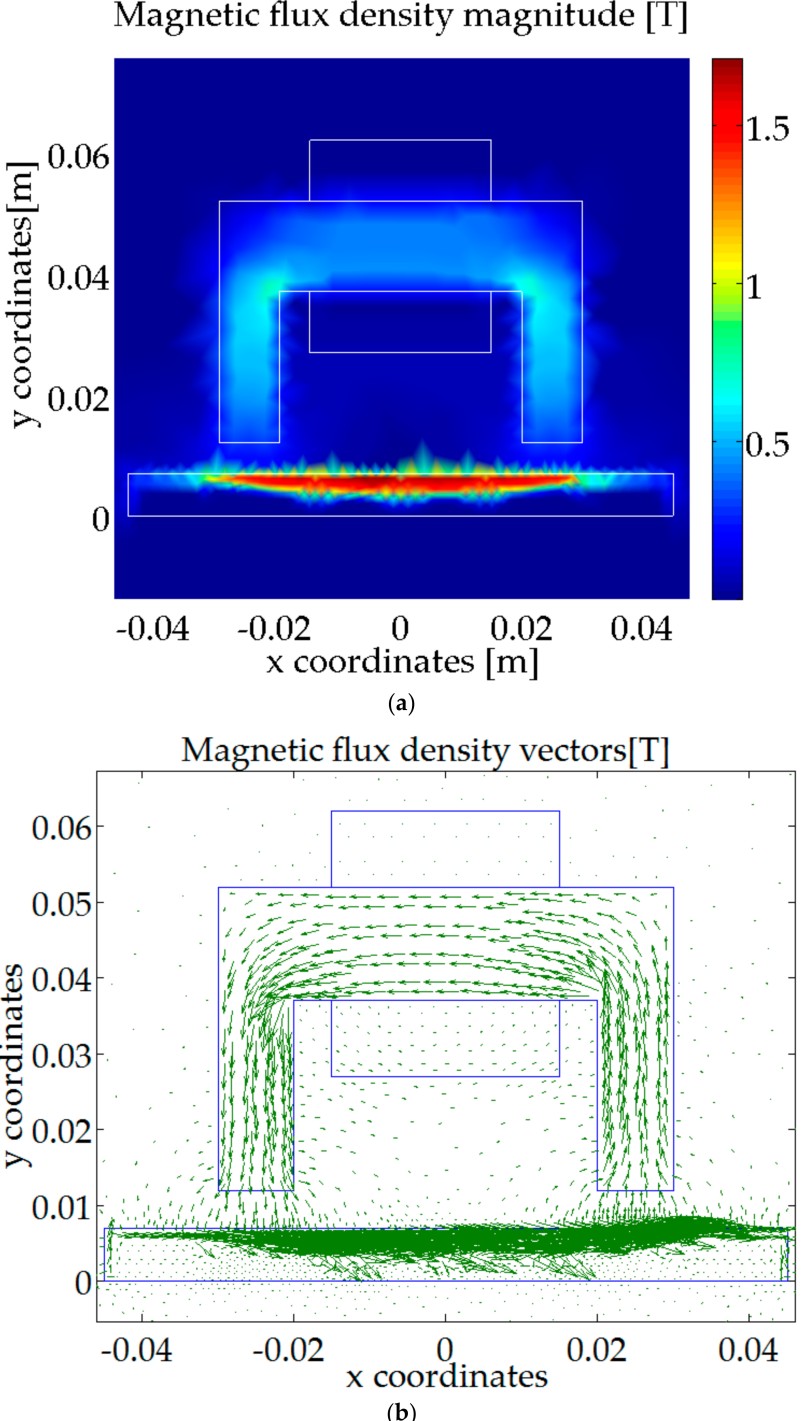

**Figure 6.** Steady state of the magnetic flux density: (**a**) spatial distribution, (**b**) vector field orientation.

According to the resistance and inductance of the winding voltage fed, Figure 7 shows that the current expressed the classical phenomena of an inductive circuit associated with a ferromagnetic core. The steady-state value of the current corresponding to the maximum value was about 78 A. The transient current behavior appeared as an image of the ferromagnetic core magnetization curve. It was observed that the reaction field of the eddy currents tilts the field line entering the secondary conductor, and it shifts the epicenters of the magnetic flux density formation above the centerlines of the plate.

Additionally, the design of the coil determines the electrical resistance and it strongly impacts the inductance of the coil, since the inductance depends on the number of square coil turns multiplied by the total magnetic permeability of the non-linear ferromagnetic core. The ratio of the resistance divided by the inductance (L/R) is the electrical time constant; it determines how fast the current can rise in the coil. Considering this rise time of the current due to the electrical time constant, the magnetic diffusion time can also impact the actuator performance.

Figures 8–10 show the transient evolution of the maximum magnitude of the induced eddy current and the magnetic force density, as well as the total magnetic force of the plate Vacofer S1 and Fe–Cu alloy materials, according to their components in the 2D $(x, y)$ Cartesian coordinates.

Figure 8 shows the time variation of the maximum values of the induced eddy current in the positions A (0, 6.5 mm), B (25 mm, 6.5 mm), and C ($-25$ mm, 6.5 mm) of the plate, as shown in Figure 2. The eddy current behavior expresses the magnetic vector potential variation according to the time. Firstly, the induced eddy current increases until an average maximum value of 5 MA/m$^2$. The second stage changes in proportion as the steady-state operation of the eddy current decreases exponentially until constant values of 0.1 MA/m$^2$. The eddy current behavior appears as an image of the magnetic flux density distribution according to the conducting plate material's magnetic permeability.

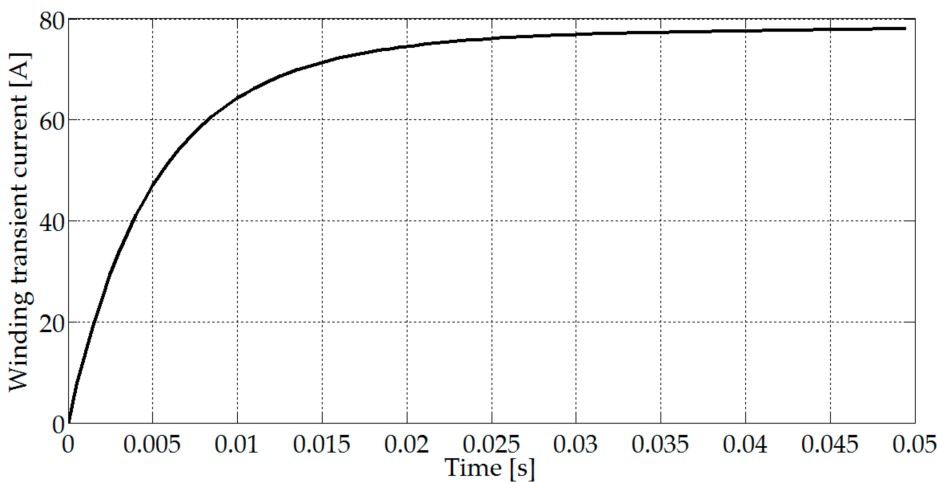

**Figure 7.** Winding transient current.

The volume magnetic force density in the non-linear magnetic and conducting materials was expressed with the Lorentz eddy current magnetic force (LZEC) based on the induced eddy currents and the magnetic flux density. The time–space distribution of the magnetic force density in the plate was similar to the induced eddy current behaviors, as shown in Figure 9 for the Vacofer S1 and Fe–Cu alloy materials. The magnetic flux in the plate preferred to pass through the shortest path because of the high magnetic permeability. Therefore, the flux passed the plate mainly in the tangential direction ($x$-direction). The magnetic forces had a perpendicular direction of the flux path; therefore, their f$y$-component was much greater than the f$x$-component. As the current rose, the electric field opposed the dissolution of the magnetic flux in the plate. Owing to the breaking of the magnetic circuit, the plate deformation commenced, and the field increased because the remaining flux was abruptly removed. After the magnetic flux was gone, the eddy currents fell back.

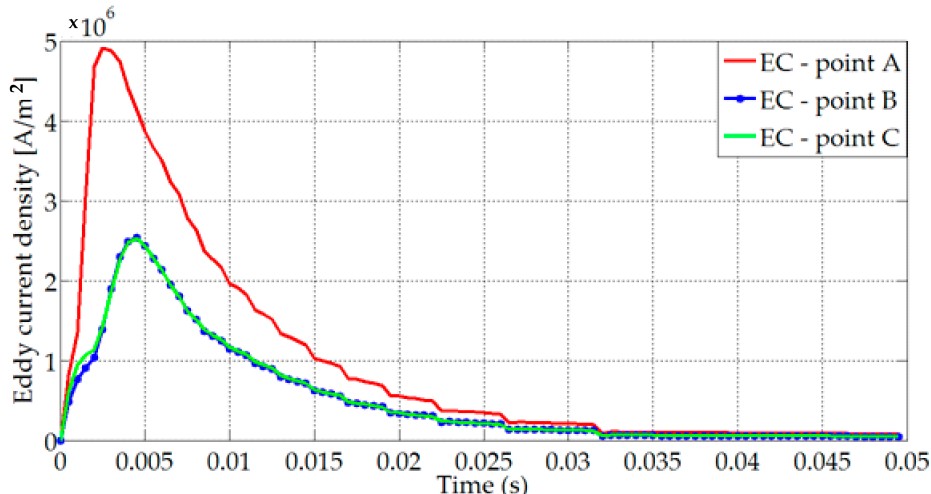

**Figure 8.** The current density induced on the plate in the non-linear magnetic material.

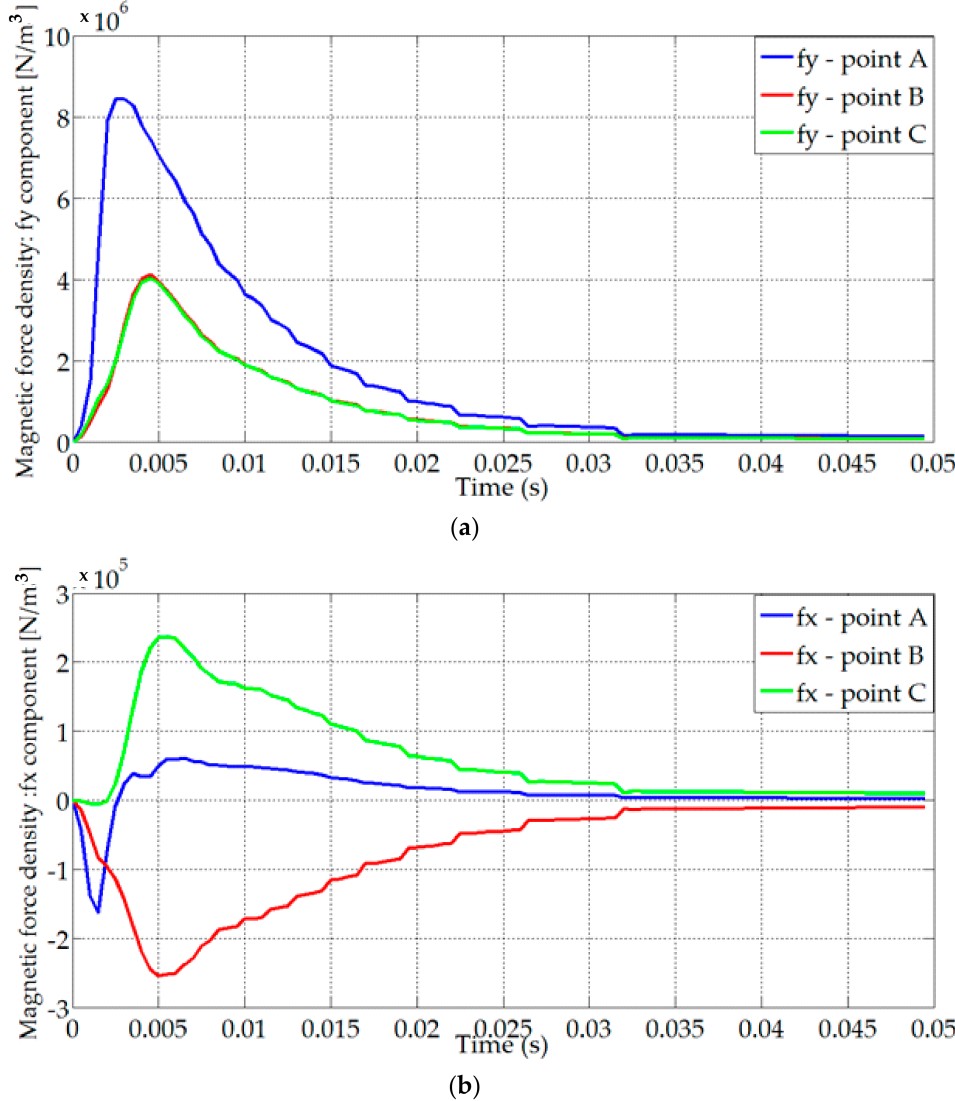

**Figure 9.** Maximum volume magnetic force density on the plate in non-linear magnetic and conductive VACOFER S1 material: (**a**) f*y*-components, (**b**) f*x*-components.

The peak values of the volume magnetic force density $fy$-components were positive and about 8.5 MN/m$^3$ and 4 MN/m$^3$, respectively, for the A and both B and C points of the plate. The average steady-state value was about 0.25 MN/m$^3$. In addition, the highest values of the magnetic force density $fx$-component were symmetric with 2.5 MN/m$^3$ value applies to both C and B points. An impulse and oscillatory behavior was noted for the $fx$-component in region A of the plate.

Figure 9 shows the time-variation components of the total magnetic for the Vacofer S1 plate material. From the result, the $y$-component of the magnetic force was dominant compared to the $x$-component. The observation of the magnetic force generated by the EMA due to increasing and decreasing eddy currents showed that the hysteresis effects were not negligible. Moreover, the generated forces were obviously proportional to the current square value [31]. The difficult parameter to define was $\mu$, since it was the permeability of the entire magnetic path, which included the magnetic material and air in the gaps.

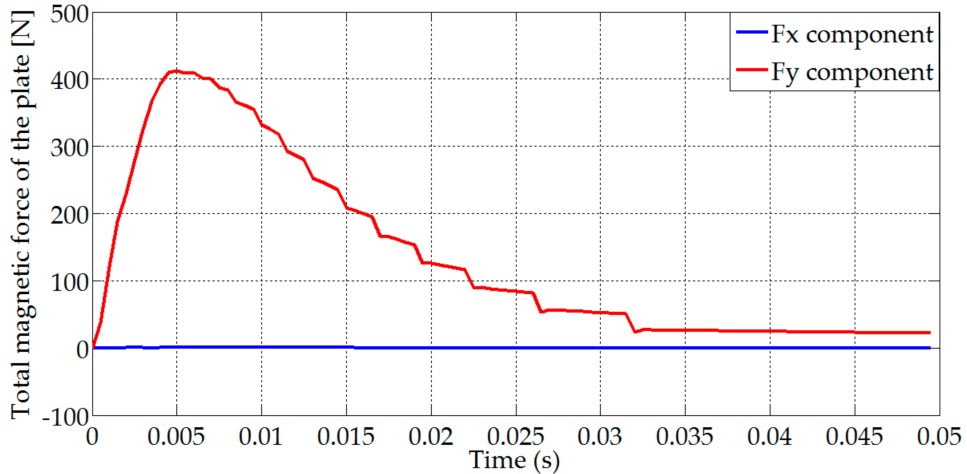

**Figure 10.** Total volume magnetic force components on the non-linear magnetic and conductive VACOFER S1 material.

The volume magnetic force density components, schematically depicted in Figure 9; Figure 10, showed a non-linear dependence of the force on the actuator's quadratic current/magnetic flux density, and a strong hyperbolic dependence of the force on the non-linear differential magnetic permeability. These non-linear force–eddy current–magnetic flux density and force–magnetic permeability relationships and the high force variation were the major reasons for the production of large forces, which allowed a design with a large deformation.

The maximum values of the surface magnetic force density components on the surrounding plate surface expressed by Equation (22) are shown in Figure 11a. In addition the trends of the time–space surface magnetic force density components are depicted in Figure 11a,b.

The electromagnetic characteristics are practically symmetric or asymmetric according to their components regarding the symmetry position of the device ($x = 0$). The $y$-component of the magnetic flux density is symmetric and unidirectional according to the positive y position, since the $x$-component is asymmetric with a change in direction from the positions $x < 0$ and $x > 0$. The ferromagnetic plate was massive and the induced eddy currents flowing in it were computed from the $z$-component of the magnetic vector potential. Although the distribution of the eddy current is not homogeneous under the plate, it remains symmetric regarding the symmetry position of the device. In contrast, the volume magnetic force density components exhibit similar patterns as the magnetic flux density components.

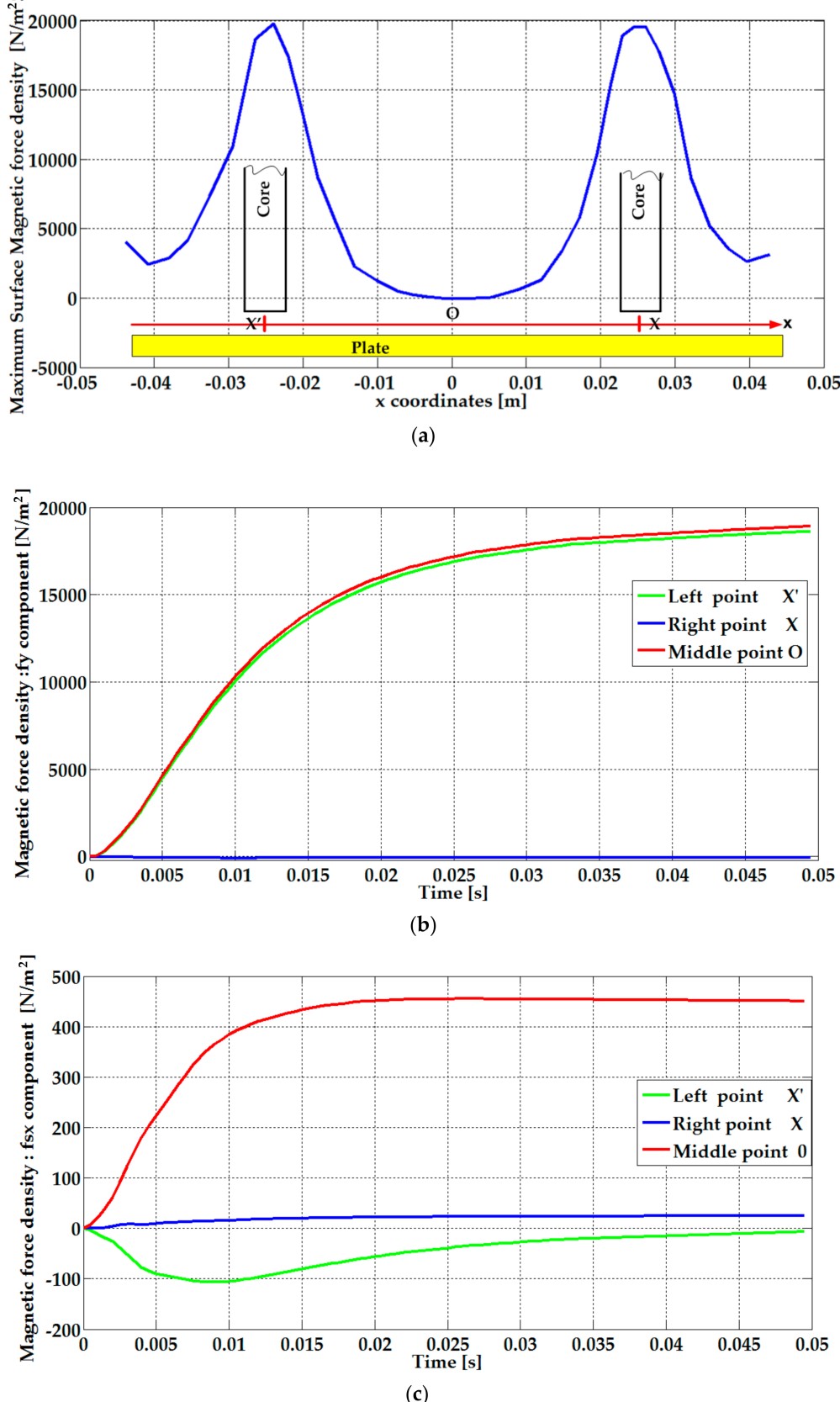

**Figure 11.** Surface magnetic force density components: (**a**) Maximum surface force, (**b**) Transient fs*y*-components, (**c**) Transient fs*x*-components.

Since most of the magnetic flux lines go through the internal plate volume combined with significant induced current density, the amplitude of the volume magnetic force density is larger than that of the surface magnetic force density one. Consequently, for the studied actuator shape, there is no significant effect on the deformation due to the surface magnetic force density.

*6.2. FEM Analyses for Structural–Mechanical Field*

In this section, we performed a transient (FEM) mechanical–structural deformation analysis of the conducting magnetic plate of the EMA under the magnetic force density excitation. The stresses in the plate were mainly provoked by their magnetic forces. The mechanical stresses were analyzed for only one selected finite element (nearest to position A), which corresponded to the highest magnetic force density and deformation strain, since similar behaviors could be observed for the other point of the plate. The created FEM tools in two dimensions were applied to calculate the stresses and the deformation according to the time variation and distribution of the magnetic force. Special attention was paid to the moments that the magnetic force densities reached a maximum, which caused significant deformations on the plate.

The structural mechanical deformation 2D FEM equation was sequentially coupled to the electromagnetic phenomena. This coupling had an advantage such that the transfer of the magnetic forces density from the Lorentz formula calculated via the electromagnetic model for the mechanical model was done with an independence between the two meshes (magnetic and mechanical). This became possible using the Lorentz eddy current magnetic force densities (LZEC) on each element of the mechanical mesh. These force densities served as input parameters for the mechanical model to determine the displacement and deformation responses of the conducting magnetic plate of the actuator. Then, these efforts were transmitted to the mechanical mesh during the resolution of the multi-level coupled models.

The transient mechanical deformation strain components according to the maximum magnetic force density obtained from the FEM of the EMF with a step voltage of 80 V in the non-linear magnetic properties (NL) are shown in Figure 12. Actually, for the driving condition at the early starting time, the plate would be pulled to the magnetic core of the *x*-components and *y*-components of the magnetic force density. Consequently, the plate strain deformation appeared mainly in the *xy*-direction due to the dominant magnetic force density components according to their time–space distribution.

The time variations of the plate deformations in the *x*-direction, *y*-direction, and *xy*-direction were mainly governed by the volume magnetic force density vectors. The plate deformations increased until a maximum value, where the $\varepsilon_{xy}$ component was lower than $\varepsilon_{yy}$, which was also lower than $\varepsilon_{xx}$. The increase in the number of deformations was due to the application of a scale of tension against this decrease in mechanical deformations. This was due to the decrease in the maximum distribution of the magnetic force density from the Lorentz formulas based on the eddy current and the magnetic state of the plate. The peak values of the deformations were $\varepsilon_{xy} = 0.375$ *u*m, $\varepsilon_{yy} = 0.15$ *u*m and $\varepsilon_{xx} = -0.04$ *u*m, for the material with the highest electrical conductivity.

The observation of the magnetic force density generated by the EMA was obviously proportional to the current square value. It was noted that accurate values of the gap distance were necessary for reliable predictions. The magnetic force characteristic strongly depended on the conductivity of the plate. The peak value of the magnetic force density components and the critical deformation increased as the conductivity of the plate increased, since the high magnetic characteristic performance had a lesser weakening influence on them.

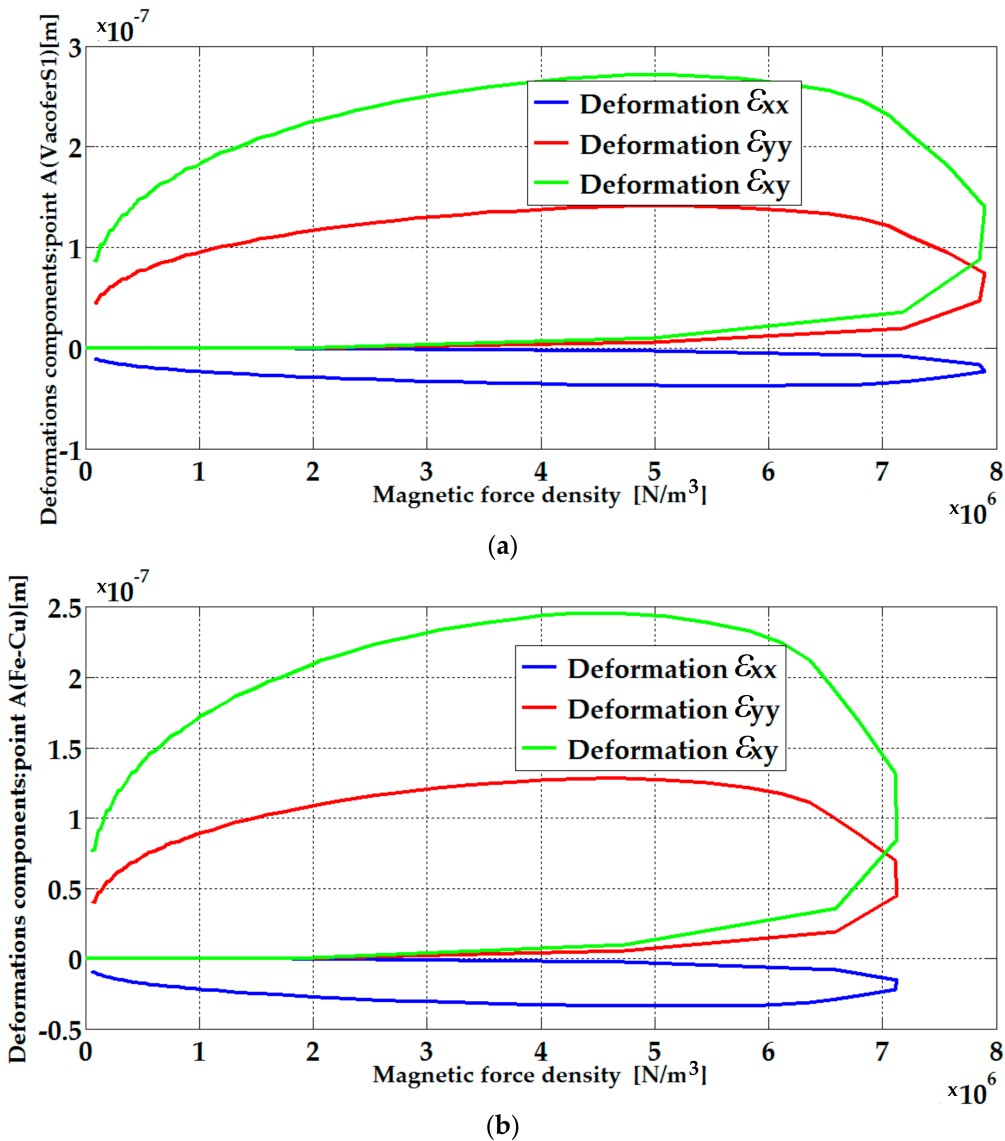

**Figure 12.** Time evolution of deformation components in the non-linear magnetic and conductive materials: (**a**) VACOFER S1, (**b**) Fe–Cu alloy.

*6.3. Analysis Parameters*

In this section, the influence of geometrical and physical parameters, such as the air-gap thickness, the supply voltage, and the electrical conductivity, on the deformation was investigated to provide useful information for the design of actuators.

The parts of the device that were studied are the most critical because, in cases of excessive deformation/stress, they can irreparably compromise the actuator operation. This part is the plate-piece. The FEM analysis of the mechanical structural deformation model allowed us to identify the most stressed areas of the previous elements whose shape was appropriately designed so as to reduce the maximum stresses and deformations. Figures 13 and 14 represent the mechanical deformation impedance, showing the *xy*-deformation component of the VacoferS1 and Fe–Cu alloy magnetic non-linear material properties according to magnetic force density at position A, with respect to the air-gap thickness and the winding step voltage magnitudes of 80 V and 120 V. Reduced air-gap thickness combined with high voltage excitation led to increased values of stress and repetitive deformations, which could exceed the fatigue limit of the plate. Therefore, the service life of the actuator will be diminished by frequent starting, or the actuator will suffer a more severe load condition for the

service that demands more frequent starting. It was noticed that accurate values of the gap distance were necessary for reliable predictions.

In addition, Figure 13 illustrates the evolution of the deformation in the *xy*-direction of the non-linear magnetic materials with different values of electrical conductivity for the Vacofer S1 and Fe–Cu alloy materials, as given in Table 2, as well as for the different values of the supply voltage of 80 V and 120 V. The maximum value of the deformation increased with the electrical conductivity, while the supplied voltage increased and, consequently, the electrical conductivity of the magnetic plate greatly impacted the behavior of the deformation according to the random distribution of the body magnetic force density. This irregular magnetic force density force pattern originated from the magnetic field direction, as well as from the non-linear magnetic material properties. When the induced eddy current density was high, the magnetic material operated in the saturation region with a high level of magnetic flux density, and the deformations due to random distribution of the body magnetic force density were much higher than when the magnetic material remained in the linear region.

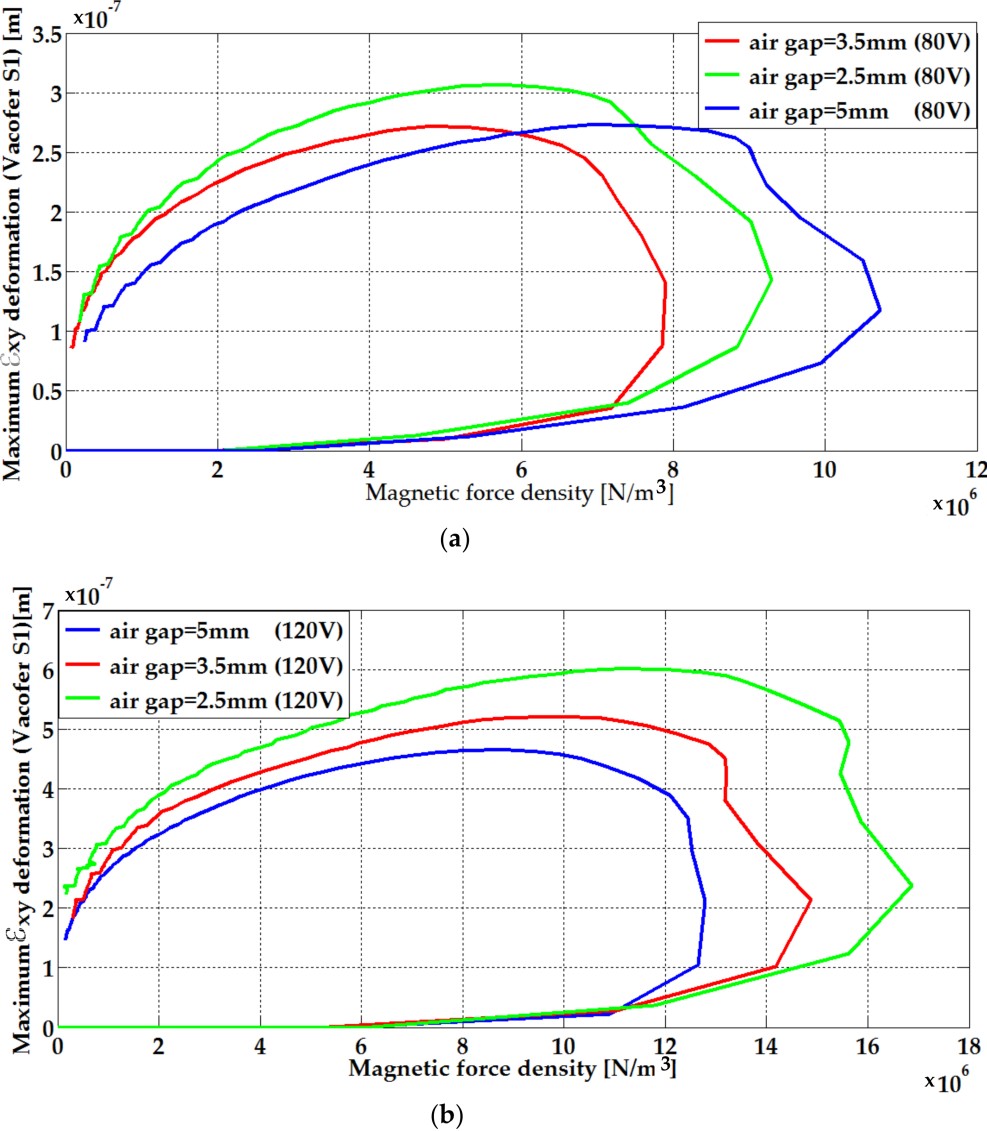

**Figure 13.** Maximum deformation according to the supplied voltage and the air-gap thickness of the Vacofer S1 magnetic non-linear and conductive material: step voltage of (**a**) 80 V, and (**b**) 120 V.

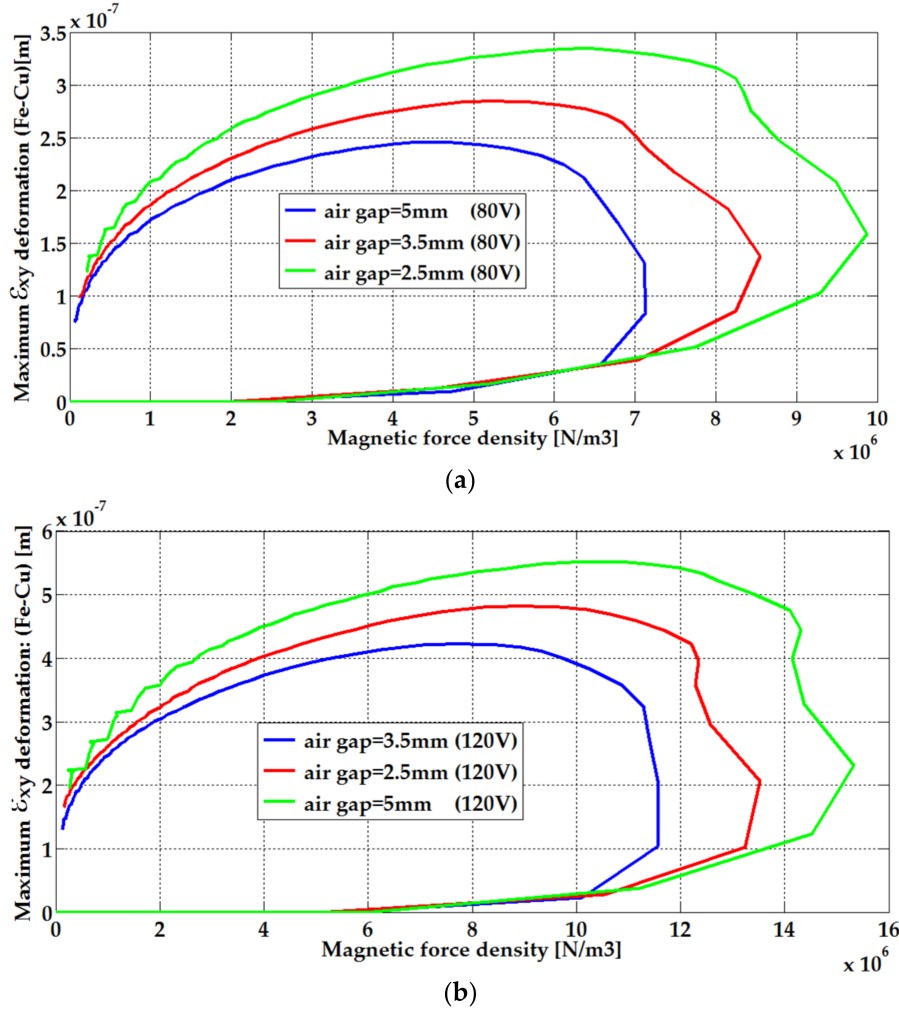

**Figure 14.** Maximum deformation according to the supplied voltage and the air-gap thickness of the Fe–Cu alloy magnetic non-linear and conductive material: step voltage of (**a**) 80 V, and (**b**) 120 V.

Table 3 indicates the peak values of the maximum deformation according to the air gap and the applied step voltage magnitude for the Vacofer S1 and Fe–Cu alloy materials.

**Table 3.** Peak values of the $\varepsilon_{xy}$ deformation according to the air-gap thickness and the voltage magnitude for the Vacofer S2 and Fe–Cu alloy materials.

| Air-Gap Thickness | Step Voltages | $\varepsilon_{xy}$ Deformation (Peak Values) [μm] | |
|---|---|---|---|
| | | **VacoferS1** | **Fe-Cu alloy** |
| 5 mm | 80 V | 0.272 | 0.246 |
| | 120 V | 0.465 | 0.423 |
| 3.5 mm | 80 V | 0.307 | 0.285 |
| | 120 V | 0.521 | 0.482 |
| 2.5 mm | 80 V | 0.363 | 0.334 |
| | 120 V | 0.602 | 0.552 |

In order to obtain a larger magnetic force, the air-gap length was chosen to be as small as possible. The plate was not only an important part of the magnetic circuit which carries the magnetic flux density, but also an important part of the medium in which the eddy currents could be induced. It could

be seen that the increasing voltage source and the electrical conductivity, as well as operating in the saturated region of the magnetic plate, resulted in a lower peak and critical deformations.

## 7. Conclusions

The paper introduced detailed multi-level coupling models dedicated to the modeling and analysis of the behavior of electromagnetic actuators. We considered transient-voltage electromagnetic fields and the structural mechanical deformation phenomena using conducting and non-linear magnetic materials. The magnetic–electric coupled field models that were solved using FEM led to magnetic force density component computations from the Lorentz eddy current magnetic force (LZEC) under transient conditions. The mechanical deformation model used the magnetic force density component as a source which was correlated with the magnetic flux density, the eddy current density, and the electrical/magnetic material properties.

An accurate analysis under different voltage sources with different air-gap thicknesses for different electric conductivity and magnetic material properties revealed an increase in magnetic force density with lower peak values of the highest deformation components.

The computed magnetic force density and structural deformation results obtained from the coupled FEM magnetic field–electric circuit and mechanical structural deformation models were qualitatively in good agreement with the models found in the scientific literature [31,32]. The structural mechanic results establishing a kind of mechanical deformation impedance can contribute to actuator design, control methodology, deformation-free and non-destructive testing, and safety, threatened by wake-induced fatigue due to repetitive deformation strain activated by a pulsed voltage source.

**Author Contributions:** The work was mainly developed by the first and second authors (F.A. and M.R.).

**Funding:** This research received no external funding.

**Conflicts of Interest:** The authors declare no conflicts of interest.

## Nomenclature

The following symbols are used in this manuscript:

| Symbol | Description |
|---|---|
| $A_z$ | z-direction component of the magnetic vector potential $\vec{A}$ |
| $E$ | Young's modulus |
| $f_{Sx}, f_{Sy}$ | Surface magnetic force density components |
| $f_{Vx}, f_{Vy}$ | Volume magnetic force density components |
| $F$ | Magnetic force density vector of the mechanical problem |
| $J_{eddy}$ | Induced eddy current density |
| $H$ | Magnetic field |
| $H_t, H_n$ | Tangential and normal magnetic field, respectively |
| $I_c(t)$ | Coil current |
| $L_{end}$ | Self-inductance of the coil |
| $N_c, N_s$ | Number of turns and the device symmetries, respectively |
| $N_{nodes}, N_{elements}^{load}$ | Number of nodes and triangular finite element of the plate |
| $R_c$ | Resistance of the coil |
| $U$ | Global displacement vector |
| $u$, v | Displacement components in $x$- and $y$-directions, respectively |
| $V_c(t)$ | Voltage of the coil |
| $\sigma(U), \varepsilon(U)$ | Mechanical stress tensor and strain tensor |
| $\sigma_{xx}, \sigma_{yy}$ | Stresses along $x$- and $y$-directions, respectively |
| $\sigma_{xy}, \sigma_{yx}$ | Stresses along $xy$- and $yx$-directions, respectively |
| $\varepsilon_{xx}, \varepsilon_{yy}, \varepsilon_{xy}$ | Deformation strain along $x$-, $y$-, and $xy$-directions, respectively |
| $\psi$, $\alpha$ | Weighting and shape functions of the FEM formulation |

| Symbol | Description |
|---|---|
| $\Omega, \Gamma$ | EMA studied domain and its surrounding boundary |
| $\Omega_{coil}, \Omega_{core,}, \Omega_{air}$ | Load plate, coil, core, and air regions, respectively |
| $\Gamma_{load}, \Omega_{load}$ | Boundary of the mechanical domain |
| $\Delta t, \theta$ | Time step and relaxation factor |
| $\sigma$ | Electric conductivity |
| $\mu(A)$ | Non-linear magnetic material permeability |
| $\mu_o, \mu_p(A)$ | Magnetic permeability of air and the plate, respectively. |
| $\nu$ | Poisson's ratio |

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
