# Peer review of "Time-Stepping FEM-Based Multi-Level Coupling of Magnetic Field–Electric Circuit and Mechanical Structural Deformation Models Dedicated to the Analysis of Electromagnetic Actuators"

_actuators, doi:10.3390/act8010022_

Round 1

Reviewer 1 Report

The paper deals with weakly coupled transient solutions of electromagnetic and structural analyses of a simple magnetic actuator. The first part of the paper is the description of the field circuit finite elements solution of 2D magnetic field inside magnetically  nonlinear and electrically conducting body. The theory of such an analysis is well known and it was given by many authors as it is referenced in the paper. Few comments stating explicitly why the Lorentz force is constant per element are suggested.

A serious doubts are connected with the second part of the work related to the mechanical behaviour of the device. It is completely incomprehensible why Authors consider the static solution - see eq.(21) and (34) given without the inertia force, while the exciting forces are transient. Besides, the boundary conditions for the displacement field are not presented in the paper. The statement l.391"...The mechanical stresses have been analyzed  for only one selected finite element because all of the plate mesh suffer very similar load conditions...." is for this reason unclear, because both surface and volumetric forces are very non-uniform. Therefore, it is not possible to estimate the quality of results presented in the last section.

Authors have not defined precisely positions of points A, B, C where the exemplary results are shown in figures, so some lack of symmetry in results  like in fig.8 cannot be commented.

The technical language must be revised, e.g. l.379 "...components of the total magnetic for the Vacofer...", l.412 "...the plate will be pulled to the magnetic core of the x-components and y-components of the magnetic force density...", use "strain" instead of "deformation" and many others.

Verify, please, correctness of equations, e.g. errors are present in: (24) - "+" instead of "-", (25) 2nd row "yx" instead of "xy".

Author Response

Dear reviewers and editor,

Thank you for your time and consideration allowed to review this article. All your questions and comments will enhance the quality of the publication. In this document, I will address your comments and explain the changes made to the paper.  The authors are still available for all other additional suggestion and question, and hope that the follow answers and improvements are satisfactory.

In addition to the review report 2,  some improvements are introduced  in the listed lines

Added  paragraph  :  from line 30 to line 36   

Added  paragraph  :  from line 48 to line 54   

Added  paragraph  :  from line 337 to line 342   

Added sentence      : line 351

Added  paragraph  :  from line 389 to line 394

Correct   paragraph :  from line 399 to line 401     (reviewer 1 suggestion)

Added  paragraph  :  from line 467 to line 471    and table.3

Added  paragraph  :  from line 487 to line 489

Added  paragraph  :  from line 490 to line 491   

Kind regards,
Dr.  M. Rachek

Reviewer 2 Report

This paper mainly presents an implementation of the electromagnetic-structural mechanic finite element method. The structural mechanical deformation equation is sequentially coupled to the electromagnetic phenomenon through the magnetic force density to get the deformations.The author conducted the review of the full research background. The conclusions can support the research process.

Author Response

Dear reviewers and editor,

Thank you for your time and consideration allowed to review this article.

Kind regards,
Dr.  M. Rachek

Reviewer 3 Report

This manuscript needs extensive editing not only for grammar but for consistency (Marrocco/Morocco), for explaining all symbols used in equations, and for clarity.

I believe there is a useful and valuable result here, that of a model that works for briefly-applied currents, but it is not introduced directly enough. For example, the key result seems to be introduced on lines 60-63 "A Lorentz force eddy current (LZEC) linear motion... was built " But, when reading the sentence it was unclear whether that result came from the literature or is what the authors did in this paper. It turns out to refer to the main result from the authors.

Other notes

-Introduction: the section between lines 46-52 "a large and particular density of the magnetic forces are undesirable" is typical of the writing. It could be "Large magnetic force density is undesirable because of..." 

-Because of its nature, this work has a large number of symbols in the equations which take a long time to define, and the symbol definition is not very thorough in this manuscript. A table with a definition of each symbol would be justified for this kind of paper.

-Is inductive heating considered in the model?

-Page 3 diagram - what is reculctivity? Many spelling errors are in the Figure 1 flow chart, compared to the rest of the manuscript.

-Figure 2, A B and C points are referenced later on in and around figure 7, so it would have helped to refer back to Figure 2.

-If possible, shared computer code in an online repository can make an algorithm paper like this have greater impact.

-The three lines 406-408 are duplicated  immediately below.

-Lines 447-448, how close does the stress come to damaging a typical plate? Is it possible to do an example calculation?

-Figures 10 and 11 do a good job showing the strong dependence of deformation on air gap for two different materials. To illustrate the point that deformation is very sensitive to the gap distance, you might also show the maximum deformation value as a function of air gap (instead of plotting vs time).

-Not enough is done to compare results to the literature and commercial code or analytical solutions (if those exist). Here is where the manuscript could be vastly improved. The conclusion does point in this direction (lines 486-488) but the sentence structure is broken  ("a great," should it be "agree?") and although a reference is given [31], no  numbers are pulled from it for comparison.

Author Response

Dear reviewers and editor,

Thank you for your time and consideration allowed to review this article. All your questions and comments will enhance the quality of the publication. In this document, I will address your comments and explain the changes made to the paper.  The authors are still available for all other additional suggestion and question, and hope that the follow answers and improvements are satisfactory.

In addition to the review report,  some improvements are introduced  in the listed lines

Added  paragraph  :  from line 30 to line 36   

Added  paragraph  :  from line 48 to line 54   

Added  paragraph  :  from line 337 to line 342   

Added sentence      : line 351

Added  paragraph  :  from line 389 to line 394

Correct   paragraph :  from line 399 to line 401     (reviewer 1 suggestion)

Added  paragraph  :  from line 467 to line 471    and table.3

Added  paragraph  :  from line 487 to line 489

Added  paragraph  :  from line 490 to line 491   

Kind regards,
Dr.  M. Rachek

Round 2

Reviewer 1 Report

The fundamental error in the paper it is the connection of transient analysis of electromagnetic force field with the steady formulation of mechanical deformation. The equations (21) and (34) do not contain inertia force, therefore, they may be applied for the static calculations only. Authors declare (line 217) "formulation structural dynamic simulation model" but definitely it is not dynamic. Besides, boundary and initial conditions in structural analysis are not given. In my opinion such a formulation has not a physical sense and it cannot be published.

Author Response

Dear reviewers and editor,

Thank you for your time and consideration allowed to review this article. All your questions and comments will enhance the quality of the publication.

The English style of the paper was improved through the English editing service of the actuators journal.

Some results and discussions are replaced to increase and improve the paper consistency.

In this document, I will address your comments and explain the changes made to the paper.  The authors are still available for all other additional suggestion and question, and hope that the follow answers and improvements are satisfactory.

The electromagnetic problem is a time dependent model. However the mechanical deformation problem is multi-static problem since the magnetic force density is time dependent.

There was confusion in the mechanic-deformation part. Certainly since the mechanical-deformation equations model is free of time (equation 21), The results of figures 10 and 11 (deformation-versus-time) lift a confusion. This is an unfortunate interpretation since the deformation depend only on the magnetic force density which the latest is realy time-dependent. The magnetic force density is computed from the time-stepping coupled electric and magnetic fields equations models. Though at, every moment we compute a magnetic force density responsible of the deformation, it is not appropriate to present this as an implicit dependence between the deformation and time.

To remove the confusion, Figures 10 and 11 are replaced by figures 10, 11 and 12 which present the variation of the xy-deformation component (point A) according to the magnetic force density. The new figures are in the same trends of lissajous curves evoking a non-destrutive evaluation from the deformation phenomena.

In addition some details related to the finite elements formulation of the mechanical-deformation equation model are introduced (equations of pages 11 and 12).

journal.

We will stay listening to your furthers comments.

Kind regards,
Dr.  M. Rachek

Reviewer 3 Report

With the revised manuscript, the flowchart is greatly improved and the table brings out the air gap results.  Some attention to grammar (principal/principle in the introduction) will clear things up further but these changes don't affect the science/technical content.

Author Response

Dear reviewers and editor,

Thank you for your time and consideration allowed to review this article.

The English style of the paper was improved through the English editing service of the actuators journal.

Some details are added to the mechanical deformation model (pages 11 and 12) .

Some results and additional discussions are replaced to complete and improve the paper consistency.

Kind regards,
Dr.  M. Rachek

Round 3

Reviewer 1 Report

1. There are two kinds of magnetic forces acting on the plate in considered actuator - volumetric forces coming from eddy currents induced in the conducting material and interacting with flux density and surface forces applied to boundary of ferromagnetic body where the magnetic flux enters the plate. Authors shows them properly eq.(31) but in further analysis the volumetric component is only taken into account. The behavior of these components in vertical 0y direction is opposed. Surface forces attract the plate to the exciter, volumetric ones push away. Their magnitudes may be estimated from data shown in the paper as follows:

Surface density equal to B^2/2miu0=25kPa for B=0.25 T see fig.5a

Volumetric density converted to equivalent surface density as fV*depth of penetration

2*5*10^6*10^-3=10 kPa see fig.7 and fig.5

The explanation of such a treatment seems necessary.

2. Authors do not introduce any constraints applied to displacements of the plate. In such a case the plate will rapidly move towards the exciter and the airgap value cannot be used as the parameter of calculations. When some points of the plate are fixed you must specify them. You must remember that net value of Fx component coming from volumetric forces may be assumed to be negligible but Fy is not equal to zero, see remarks above. The surface forces are always far from zero.

3. The usage of the static approach in transient mechanical calculations of deformation can be accepted when the net force applied amounts to zero and simultaneously the stiffness of the system prevails its inertia. Looking on dimensions of analysed actuator probably it can be used but in the case only when the plate is fixed somehow. Few words of explanations supporting this are expected.

If plate is free to move as a rigid body you must always take its mass and acceleration into account.

General remark: It is still not clear which physical phenomena are considered in the paper

Author Response

Dear reviewers and editor,

Thank you again for your great interest of the paper, and for your important asked questions in order to increase the paper quality.

Following the previous questions, please find the responses detailed through   the new  explanatory paragraphs  introduced in the paper. Also additional results and discussions are replaced to increase and improve the paper consistency.

Questions 1 and 2

1. There are two kinds of magnetic forces acting on the plate in considered actuator - volumetric forces coming from eddy currents induced in the conducting material and interacting with flux density and surface forces applied to boundary of ferromagnetic body where the magnetic flux enters the plate. Authors shows them properly eq.(31) but in further analysis the volumetric component is only taken into account. The behavior of these components in vertical 0y direction is opposed. Surface forces attract the plate to the exciter, volumetric ones push away. Their magnitudes may be estimated from data shown in the paper as follows:

Surface density equal to B^2/2miu0=25kPa for B=0.25 T see fig.5a

Volumetric density converted to equivalent surface density as fV*depth of penetration

2*5*10^6*10^-3=10 kPa see fig.7 and fig.5

The explanation of such a treatment seems necessary.

2. Authors do not introduce any constraints applied to displacements of the plate. In such a case the plate will rapidly move towards the exciter and the airgap value cannot be used as the parameter of calculations. When some points of the plate are fixed you must specify them. You must remember that net value of Fx component coming from volumetric forces may be assumed to be negligible but Fy is not equal to zero, see remarks above. The surface forces are always far from zero.

The surface force is about 10 times lower than the volume force density.

The Amperian representation with current density sources is one of the commonly used formulations to explain the two terms of the magnetic force density. One corresponds to a volume contribution and  the other to a surface one. These terms come directly from the equivalent volume current sources                                                 and surface current sources  associated to the magnetized state of the body. Both current density are linked with volume and surface magnetic force densities respectively.

The detailed formulations are added in section 4 (pages 8, 9 and 10) through parapgraphs and formulas (19), (22) and (23).

The quantitative values and trends of the surface magnetic force density is also incorporated in section 6.1  (pages 19 and 20) as figures 11.(a,b,c)

The comparison of the volume and surface magnetic force density shows that the volume magnetic force is widely higher than the surface one. This is mainly due to the plate electrical conductivity which leads to increased induced eddy currents. In addition the magnetic flux density magnitude increase according to magnetic non-linear behavior of the plate. 

For non conducting plate, the force magnetic density behavior will be the reverse. 

The boundary conditions applied as displacement constraints are detailed in page 14 through the  Figure 4. The plate is considered as clamped-clamped in both sides.

3. The usage of the static approach in transient mechanical calculations of deformation can be accepted when the net force applied amounts to zero and simultaneously the stiffness of the system prevails its inertia. Looking on dimensions of analysed actuator probably it can be used but in the case only when the plate is fixed somehow. Few words of explanations supporting this are expected.

If plate is free to move as a rigid body you must always take its mass and acceleration into account.

At this stage, only the static mechanical-deformation is considered since the deformation are so-small. The inclusion of the transient term due to the dynamic inertia in the mechanical deformation model concern the further development of the paper. Harmonic analysis of eigenmodes are also projected.

General remark: It is still not clear which physical phenomena are considered in the paper.

The transient strongly coupled of the electric circuit equation and magnetic field equation is considered for the finite element electromagnetic modeling of the actuator. This lead to time-space distribution of the magnetic flux density, eddy current density and magnetic force density in the plate region. 

Using the magnetic flux density as source term in the considered static mechanical equilibrium equation lead to compute the plate deformation. For the considered small deformation and since the magnetostriction phenomena is neglected, there is no strong coupling  relation between the magnetic field and the deformation of the body. No-deformed or moved mesh of the body is considered.

Finally the paper is positioned in the research area of mechanical behavior non-destructive evaluation through the concept of mechanical-deformation impedance.

The authors are still available for all other additional suggestion and question, and hope that the follow answers and improvements are satisfactory.
